# The alternative sigma factor σ$^X$ mediates competence shut-off at the cell pole in *Streptococcus pneumoniae*

Calum HG Johnston[1,2], Anne-Lise Soulet[1,2], Matthieu Bergé[1,2,3], Marc Prudhomme[1,2], David De Lemos[1,2], Patrice Polard[1,2]*

[1]Laboratoire de Microbiologie et Génétique Moléculaires (LMGM ; UMR5100), Centre de Biologie Intégrative (CBI), Centre Nationale de la Recherche Scientifique (CNRS), Toulouse, France; [2]Université Paul Sabatier (Toulouse III), Toulouse, France; [3]Dept. Microbiology and Molecular Medicine, Institute of Genetics & Genomics in Geneva (iGE3), Faculty of Medicine, University of Geneva, Geneva, Switzerland

**Abstract** Competence is a widespread bacterial differentiation program driving antibiotic resistance and virulence in many pathogens. Here, we studied the spatiotemporal localization dynamics of the key regulators that master the two intertwined and transient transcription waves defining competence in *Streptococcus pneumoniae*. The first wave relies on the stress-inducible phosphorelay between ComD and ComE proteins, and the second on the alternative sigma factor σ$^X$, which directs the expression of the DprA protein that turns off competence through interaction with phosphorylated ComE. We found that ComD, σ$^X$ and DprA stably co-localize at one pole in competent cells, with σ$^X$ physically conveying DprA next to ComD. Through this polar DprA targeting function, σ$^X$ mediates the timely shut-off of the pneumococcal competence cycle, preserving cell fitness. Altogether, this study unveils an unprecedented role for a transcription σ factor in spatially coordinating the negative feedback loop of its own genetic circuit.

**\*For correspondence:**
patrice.polard@univ-tlse3.fr

**Competing interests:** The authors declare that no competing interests exist.

## Introduction

In bacteria, sigma (σ) factors are essential transcription effectors that direct the RNA polymerase to and activate RNA synthesis at specific genes promoters. All bacterial species encode a single, highly conserved σ factor that drives the expression of house-keeping genes essential for vegetative growth and cell homeostasis. In addition, many bacteria encode a variable set of alternative σ factors that control specific regulons, providing appropriate properties to the cells in response to various stimuli. These alternative σ factors play pivotal roles in the multifaceted lifestyles of bacteria. They trigger specific developmental programs, such as sporulation or biofilm formation, as well as adapted responses to multiple types of stress and virulence in some pathogenic species (*Kazmierczak et al., 2005*). How these alternative σ factors are activated in the cell has been extensively studied, revealing multiple mechanisms underlying their finely tuned regulation (*Österberg et al., 2011*). However, how these mechanisms are orchestrated spatiotemporally within the cell remains poorly understood.

The human pathogen *Streptococcus pneumoniae* (the pneumococcus) possesses a unique alternative σ factor σ$^X$ (*Lee and Morrison, 1999*). It is key to the regulatory circuit controlling the transient differentiation state of competence. Pneumococcal competence is induced in response to multiple types of stresses, such as antibiotic exposure (*Prudhomme et al., 2006*; *Slager et al., 2014*). This induction modifies the transcriptional expression of up to 17% of genes (*Aprianto et al., 2018*; *Dagkessamanskaia et al., 2004*; *Peterson et al., 2004*; *Slager et al., 2019*). Competence is a key feature in the lifestyle of pneumococci as it promotes natural transformation, a horizontal gene

transfer process widespread in bacteria that facilitates adaptation by acquisition of new genetic traits (*Johnston et al., 2014*). In addition, pneumococcal competence development provides the cells with the ability to attack non-competent cells, a scavenging property defined as fratricide (*Claverys and Håvarstein, 2007*), is involved in biofilm formation (*Aggarwal et al., 2018*; *Vidal et al., 2013*) and virulence (*Johnston et al., 2018*; *Lin et al., 2016*; *Lin and Lau, 2019*; *Zhu et al., 2015*).

Pneumococcal competence induction is primarily regulated by a positive feedback loop involving the genes encoded by the *comAB* and *comCDE* operons (*Figure 1A*). The *comC* gene codes for a peptide pheromone coordinating competence development within the growing cell population. This peptide, accordingly named CSP (Competence Stimulating Peptide), is secreted by the dedicated ComAB transporter (*Hui et al., 1995*). After export, it promotes autophosphorylation of the membrane-bound two-component system histidine kinase ComD, which in turn phosphorylates its cognate intracellular response regulator ComE (*Figure 1A*). Phosphorylated ComE (ComE~P) specifically induces the expression of 25 genes, which include the *comAB* and *comCDE* operons, generating a positive feedback loop that controls competence development. Conversely, unphosphorylated ComE acts as repressor of its own regulon, the expression of which is thus modulated by the ComE/ComE~P ratio (*Martin et al., 2013*). The ComE regulon includes two identical genes encoding $\sigma^X$, named *comX1* and *comX2* (*Lee and Morrison, 1999*). The $\sigma^X$ regulon comprises ~60 genes, with ~20 involved in natural transformation (*Claverys et al., 2006*; *Peterson et al., 2004*), five in fratricide (*Claverys and Håvarstein, 2007*) but the majority having undefined roles. The reason why the $\sigma^X$-encoding gene is duplicated is unknown, the inactivation of one of them having no impact on transformation (*Lee and Morrison, 1999*). To fully activate transcription, $\sigma^X$ needs to be assisted by ComW, another protein whose production is controlled by ComE~P (*Luo et al., 2004*). ComW is proposed to help $\sigma^X$ association with the RNA polymerase at promoter sequences presenting the consensual 8 bp *cin* box motif (*Peterson et al., 2004*; *Sung and Morrison, 2005*). Altogether, ComE~P and $\sigma^X$ trigger two successive waves of competence (*com*) gene transcription, commonly referred to as early and late, respectively. Importantly, competence shut-off is mediated by the late *com* protein DprA (*Mirouze et al., 2013*; *Weng et al., 2013*), which directly interacts with ComE~P to turn-off ComE~P-dependent transcription (*Mirouze et al., 2013*). In addition to defining the negative feedback loop of the pneumococcal competence regulatory circuit, DprA also plays a crucial, conserved role in transformation by mediating RecA polymerization onto transforming ssDNA to facilitate homologous recombination (*Mortier-Barrière et al., 2007*; *Quevillon-Cheruel et al., 2012*; *Figure 1A*). Over 8000 molecules of DprA are produced per competent cell (*Mirouze et al., 2013*). Although only ~300–600 molecules are required for optimal transformation, full expression of *dprA* is required for optimal competence shut-off (*Johnston et al., 2018*). Uncontrolled competence induction in cells lacking DprA results in a large in vitro growth defect, and high cellular levels of DprA thus maintain the fitness of the competent population and of resulting transformants (*Johnston et al., 2018*). In addition, inactivation of *dprA* was shown to be highly detrimental for development of pneumococcal infection, dependent on the ability of cells to develop competence (*Lin and Lau, 2019*; *Zhu et al., 2015*). Together, these studies showed that the DprA-mediated shut-off of pneumococcal competence is key for pneumococcal cell fitness.

A hallmark of pneumococcal competence is its tight temporal window, which lasts less than 30 minutes in actively dividing cells (*Alloing et al., 1998*; *Håvarstein et al., 1995*). How this regulation is coordinated within the cell remains unknown. Here, we studied the choreography of pneumococcal competence induction and shut-off at the single cell level by tracking the spatiotemporal localization of the main effectors of these processes, DprA, $\sigma^X$, ComW, ComD, ComE and exogenous CSP. Remarkably, DprA, $\sigma^X$, ComD, CSP and to some extent ComE were found to colocalize at a single-cell pole during competence. This study revealed that the entire pneumococcal competence cycle occurs at cell pole, from its induction triggered by ComD, ComE, and CSP to its shut-off mediated by DprA and assisted by $\sigma^X$. In this regulatory mechanism, $\sigma^X$ is found to exert an unprecedented role for a σ factor. In addition to directing the transcription of the *dprA* gene, $\sigma^X$ associates with and anchors DprA at the same cellular pole where ComD and CSP are located, allowing this repressor to interact with newly activated ComE~P and promoting timely extinction of the whole transcriptional regulatory circuit of competence.

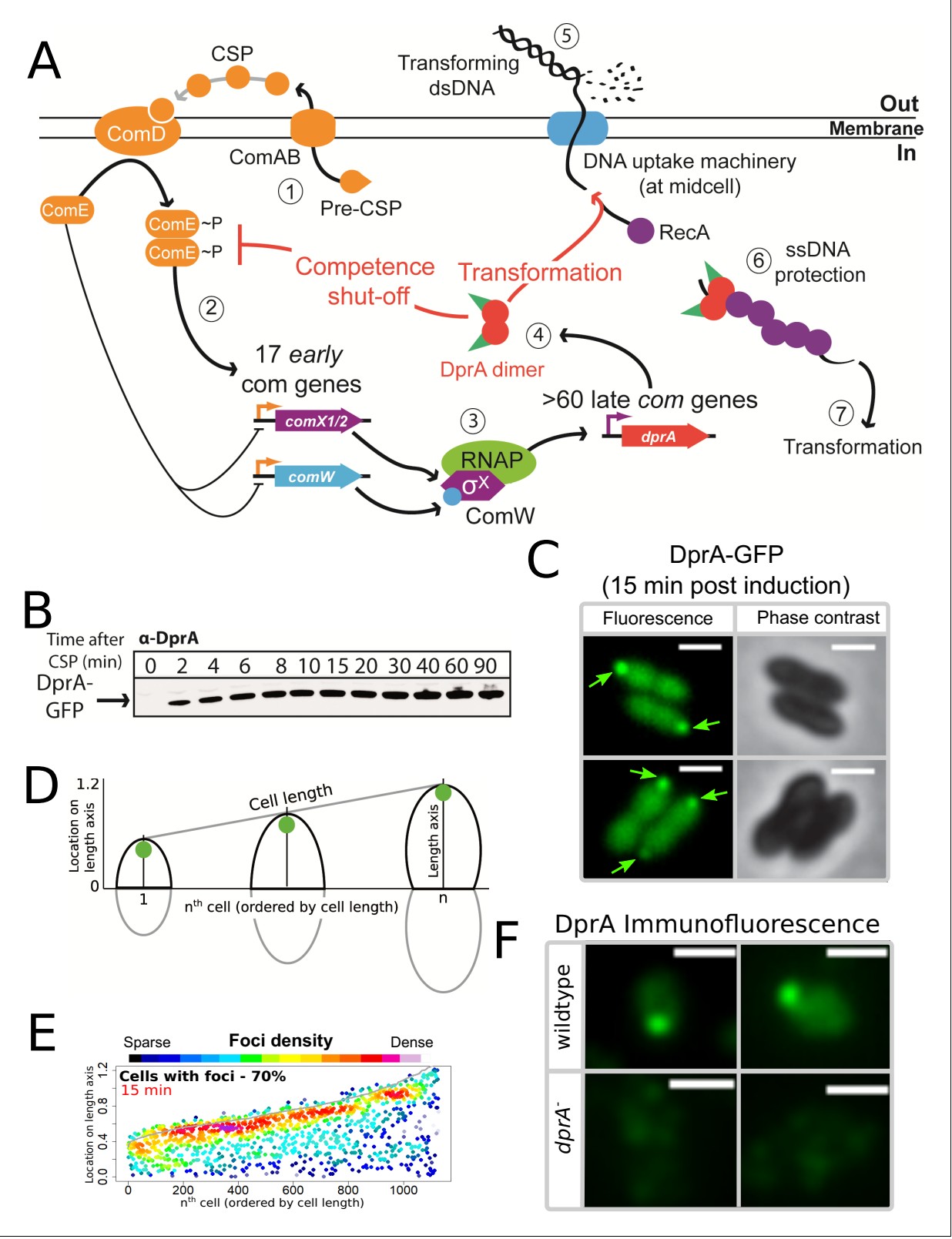

**Figure 1.** DprA: Localization and roles in competence and transformation. (**A**) (1) Pre-CSP is exported and matured by the ComAB transporter, and then promotes phosphorylation of the histidine kinase ComD. (2) ComD transphosphorylates ComE, which then stimulates the expression of 17 early *com* genes, including two copies of *comX*. (3) These encode an alternative sigma factor σ^X, which controls late *com* genes including *dprA*. (4) DprA dimers load RecA onto ssDNA to mediate transformation and interact with ComE~P to shut-off competence. (5) Transforming DNA is internalized in

*Figure 1 continued on next page*

*Figure 1 continued*

single strand form and is protected from degradation by DprA and RecA (6), which then mediate transformation (7). Orange arrows, early *com* promoters; purple arrows, late *com* promoters. (B) Western blot tracking cellular levels of DprA-GFP after competence induction in strain R3728. α-DprA antibody used. (C) Sample fluorescence microscopy images of R3728 strain producing DprA-GFP 15 min after competence induction. Scale bars, 1 µm. (D) Schematic representation of focus density maps with half cells represented as vertical lines in ascending size order and localization of foci represented along the length axis of each half cell. Black half-cells represent those presented, and grey those not presented. (E) DprA-GFP accumulates at the cell poles during competence. 1290 cells and 1128 foci analyzed. (F) Sample immunofluorescence microscopy images of a strain producing wild-type DprA (R1502; wildtype) and a strain lacking DprA (R2018; *dprA⁻*) fixed 15 min after competence induction and probed using anti-DprA antibodies. Scale bars, 1 µm.

The online version of this article includes the following figure supplement(s) for figure 1:

**Figure supplement 1.** Validation of DprA-GFP activity.

## Results

### DprA displays a polar localization in competent cells, which correlates with competence shut-off

To investigate the localization of DprA during competence in live cells, we used a fluorescent fusion protein DprA-GFP, produced from the native *dprA* locus (*Figure 1—figure supplement 1A*). DprA-GFP was synthesized during competence and remained stable up to 90 min after induction (*Figure 1B*), similarly to wild-type DprA (*Mirouze et al., 2013*), without degradation (*Figure 1—figure supplement 1B*). A strain possessing DprA-GFP was almost fully functional in transformation (*Figure 1—figure supplement 1C*), but partially altered in competence shut-off (*Figure 1—figure supplement 1D*). In pneumococcal cells induced with exogenous CSP, the peak of competence induction occurs 15–20 min after induction. 15 min after competence induction, DprA-GFP showed a diffuse cytoplasmic localization, punctuated by discrete foci of varying intensity (*Figure 1C*). The distribution and localization of DprA-GFP foci was analyzed by MicrobeJ (*Ducret et al., 2016*), with results presented as focus density maps ordered by cell length. Spots represent the localization of a DprA-GFP focus on a representative half pneumococcal cell, while spot color represents density of foci at a particular cellular location. DprA-GFP foci were present in 70% of cells, predominantly at a single cell pole (*Figure 1DE*). To ascertain whether the polar localization of DprA-GFP was due to the GFP tag or represented functional localization, we carried out immunofluorescence microscopy with competent cells possessing wild-type DprA using α–DprA antibodies. Results showed that native DprA exhibited a similar accumulation pattern as the DprA-GFP foci upon competence induction (*Figure 1F*).

Next, to explore the relation between these foci and the dual role of DprA in transformation and competence regulation, we investigated focus formation in cells possessing a previously published mutation in DprA impairing its dimerization (DprA$^{AR}$) (*Quevillon-Cheruel et al., 2012*). This mutant strongly affected both transformation and competence shut-off (*Mirouze et al., 2013*; *Quevillon-Cheruel et al., 2012*). Results showed that 15 min after competence induction, DprA$^{AR}$-GFP did not form foci, despite being produced at wild-type levels (*Figure 1—figure supplement 1E*). DprA thus accumulates at the cell pole during competence, dependent on its ability to dimerize. To explore whether the polar foci of DprA were involved in its role in transformation, or competence shut-off, we began by investigating how a *dprA$^{QNQ}$* mutation, specifically abrogating the interaction between DprA and RecA and thus affecting transformation (*Quevillon-Cheruel et al., 2012*), affected the localization of DprA-GFP. Fifteen minutes after competence induction, the DprA$^{QNQ}$-GFP mutant formed polar foci at wild-type levels (*Figure 1—figure supplement 1F*). In addition, the inactivation of *comEC*, encoding for an essential protein of the transmembrane DNA entry pore (*Pestova and Morrison, 1998*), or *recA*, encoding the recombinase with which DprA interacts during transformation (*Mortier-Barrière et al., 2007*), did not alter the frequency or localization of DprA-GFP foci (*Figure 1—figure supplement 1GH*). Altogether, these results suggested that the polar foci of DprA-GFP may not be related to the DNA entry or recombination steps of transformation but could be linked to competence shut-off.

We recently reported that optimal competence shut-off relies on the maximal cellular concentration of DprA (~8000 molecules), while wild-type transformation required only around 1/10th of these molecules (*Johnston et al., 2018*). This conclusion was obtained by expressing *dprA* under the

control of the IPTG-inducible P$_{lac}$ promoter (CEP$_{lac}$-dprA), which enables the modulation of the cellular concentration of DprA by varying IPTG concentration in the growth medium. Here, we reproduced these experiments with the DprA-GFP fusion, to test whether its concentration correlates with the formation of polar foci in competent cells. The expression, transformation, and competence profiles of a dprA⁻ mutant strain harboring the ectopic CEP$_{lac}$-dprA-gfp construct in varying concentrations of IPTG (*Figure 2—figure supplement 1*) were equivalent to those reported previously for CEP$_{lac}$-dprA (*Johnston et al., 2018*). Notably, a steady decrease in DprA-GFP foci was observed as IPTG was reduced (*Figure 2AB*). When comparing the cellular localization of DprA-GFP foci, a sharp reduction in the proportion of polar foci was observed as IPTG was reduced, with most of the remaining foci observed at midcell and appearing weaker in intensity (*Figure 2AC*). This shift correlated with a progressive loss of competence shut-off (*Figure 2—figure supplement 1C*), presenting a strong link between the presence of polar DprA-GFP foci and the shut-off of pneumococcal competence. Altogether, these results strongly support the notion that the polar foci of DprA-GFP represent the subcellular site where DprA mediates competence shut-off.

Finally, to further explore the temporal dynamics of these DprA-GFP foci, their distribution within competent cells was analyzed over the competence period and beyond. The results are presented in *Figure 3A* as focus density maps ordered by cell length. The number of cells with foci, as well as their intensity, was found to increase gradually to reach a maximum of 74% at 30 min after competence induction (*Figure 3A*), with the majority of cells possessing a single focus that persisted long after induction (*Figure 3B*). Notably, the DprA-GFP foci localization pattern rapidly evolved from a central position to a single cell pole (*Figure 3C*). DprA-GFP foci were not observed in a particular cell type, with polar foci found in small, large or constricted cells throughout competence (*Figure 3C*). Finally, tracking DprA-GFP foci formed in the cells after 10 min of competence induction by time-lapse microscopy showed that once generated, they remained static over 20 min (*Figure 1* and *Figure 1—figure supplement 1IJ*, *Video 1*). In conclusion, DprA-GFP forms discrete and static polar foci during competence, with most cells possessing a single focus. This polar localization of DprA correlates with its regulatory role in competent shut-off.

## The polar localization of DprA-GFP requires induction of the late *com* genes

Transcriptional expression of *dprA* is only detected during competence (*Aprianto et al., 2018*; *Dagkessamanskaia et al., 2004*; *Peterson et al., 2004*). To explore whether a competence-specific factor was required for the formation of polar DprA-GFP foci during competence, *dprA-gfp* was ectopically expressed from a promoter inducible by the BIP peptide in *dprA⁻* cells. This BIP-derived induction mimics rapid, strong induction by CSP during competence (*Johnston et al., 2016*). These cells were found to produce stable DprA-GFP after exposure to BIP, and upon addition of CSP to the growth medium, to transform at wild-type levels and to partially shut-off competence (*Figure 4—figure supplement 1*). However, BIP-induced production of DprA-GFP in the absence of CSP resulted in the formation of weak, barely detectable polar foci in only 10% of non-competent cells (*Figure 4A, B*). In comparison, 47% of cells producing DprA-GFP during competence formed bright polar foci (*Figure 4A, B*). This stark increase in DprA-GFP foci showed that a competence-specific factor was crucial for their formation and anchoring at the cell pole. To explore whether the competence-specific factor needed for the polar localization of DprA-GFP in competent cells was part of the early or late *com* regulons, we generated two constructs allowing us to artificially control DprA-GFP expression and either one or the other of these two connected regulons (*Figure 4C, D* and Materials and methods). Observation of DprA-GFP in conditions where only late *com* genes were induced revealed the presence of polar foci at wildtype levels (*Figure 4E, F*). Conversely, no foci were observed when DprA-GFP was ectopically expressed with only the early *com* genes (*Figure 4E*), showing that late *com* regulon expression was required for the polar accumulation of DprA-GFP.

## The polar localization of DprA-GFP depends on the alternative sigma factor σ$^X$

The late *com* regulon is comprised of 62 genes, organized in 18 operons (*Claverys et al., 2006*; *Dagkessamanskaia et al., 2004*; *Peterson et al., 2004*). To identify the hypothetical late *com* gene

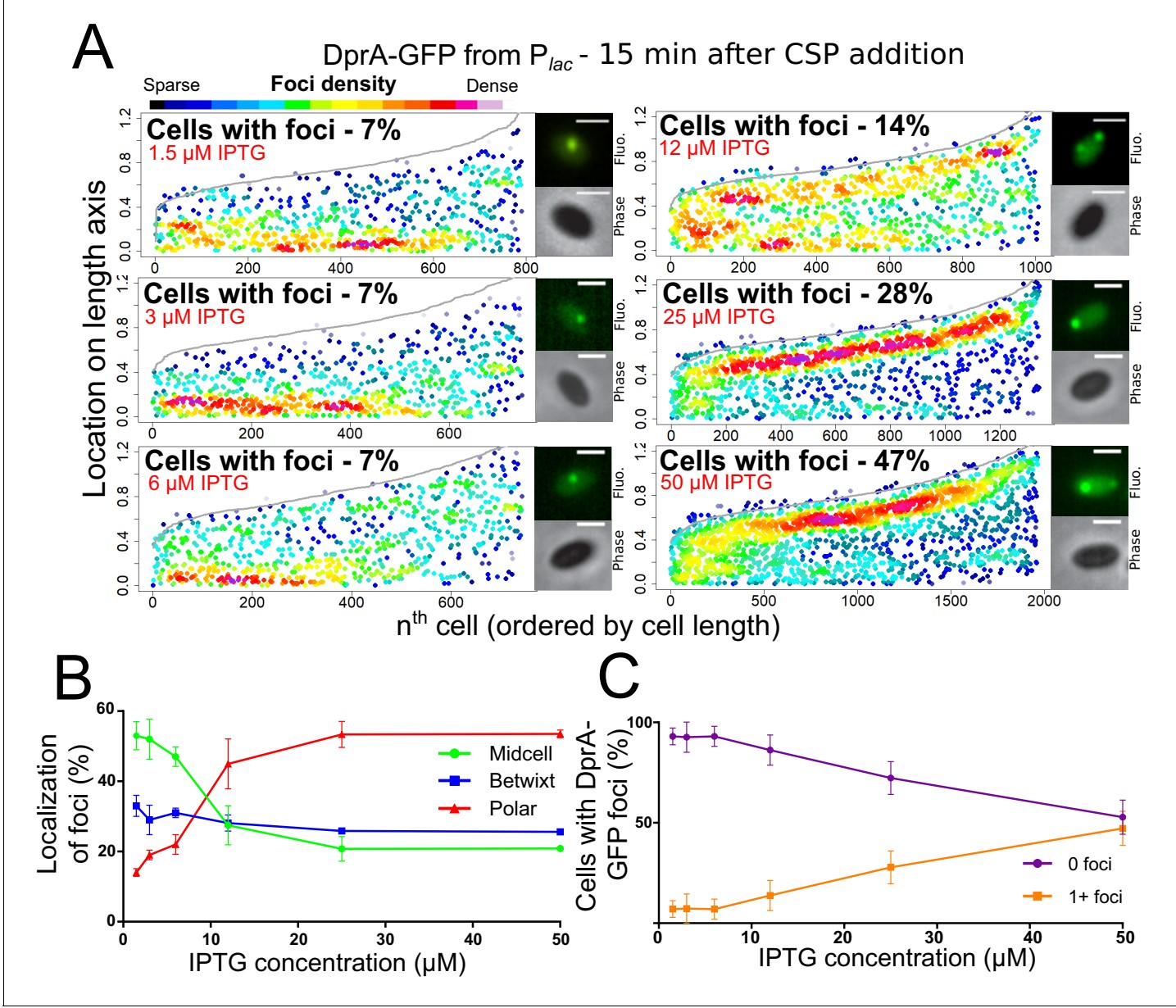

**Figure 2.** Reducing cellular DprA-GFP levels results in loss of polar accumulation and competence shut-off. (A) Focus density maps of DprA-GFP foci at different cellular levels during competence. *CEP_{lac}-dprA-gfp* from strain R4262. Cellular DprA-GFP levels were controlled by growing cells in a gradient of IPTG. 1.5 µM IPTG, 11267 cells and 791 foci analyzed; 3 µM IPTG, 10623 cells and 748 foci analyzed; 6 µM IPTG, 10603 cells and 743 foci analyzed; 12.5 µM IPTG, 6985 cells and 1010 foci analyzed; 25 µM IPTG, 2945 cells and 1345 foci analyzed; 50 µM IPTG, 3678 cells and 1964 foci analyzed. Sample microscopy images of strain R4262 in varying IPTG concentrations. Scale bars, 1 µm. (B) Reducing cellular levels of DprA-GFP reduces the number of cells with foci. Error bars represent triplicate repeats. (C) Reducing cellular levels of DprA-GFP results specifically in loss of polar foci. Error bars represent triplicate repeats.

The online version of this article includes the following figure supplement(s) for figure 2:

**Figure supplement 1.** Validation of *CEP_{lac}-dprA-gfp* activity.

product needed for DprA localization at the cell pole, the *cin* boxes that define the late *com* promoters were individually mutated, generating a panel of 18 mutant strains, each lacking the ability to induce a specific late *com* operon. The inactivation was validated by comparing transformation efficiency in three strains, where *cin* box inactivation mirrored gene knockout levels (*Supplementary file 1*). Visualization of the red fluorescent fusion DprA-mKate2 showed that in all

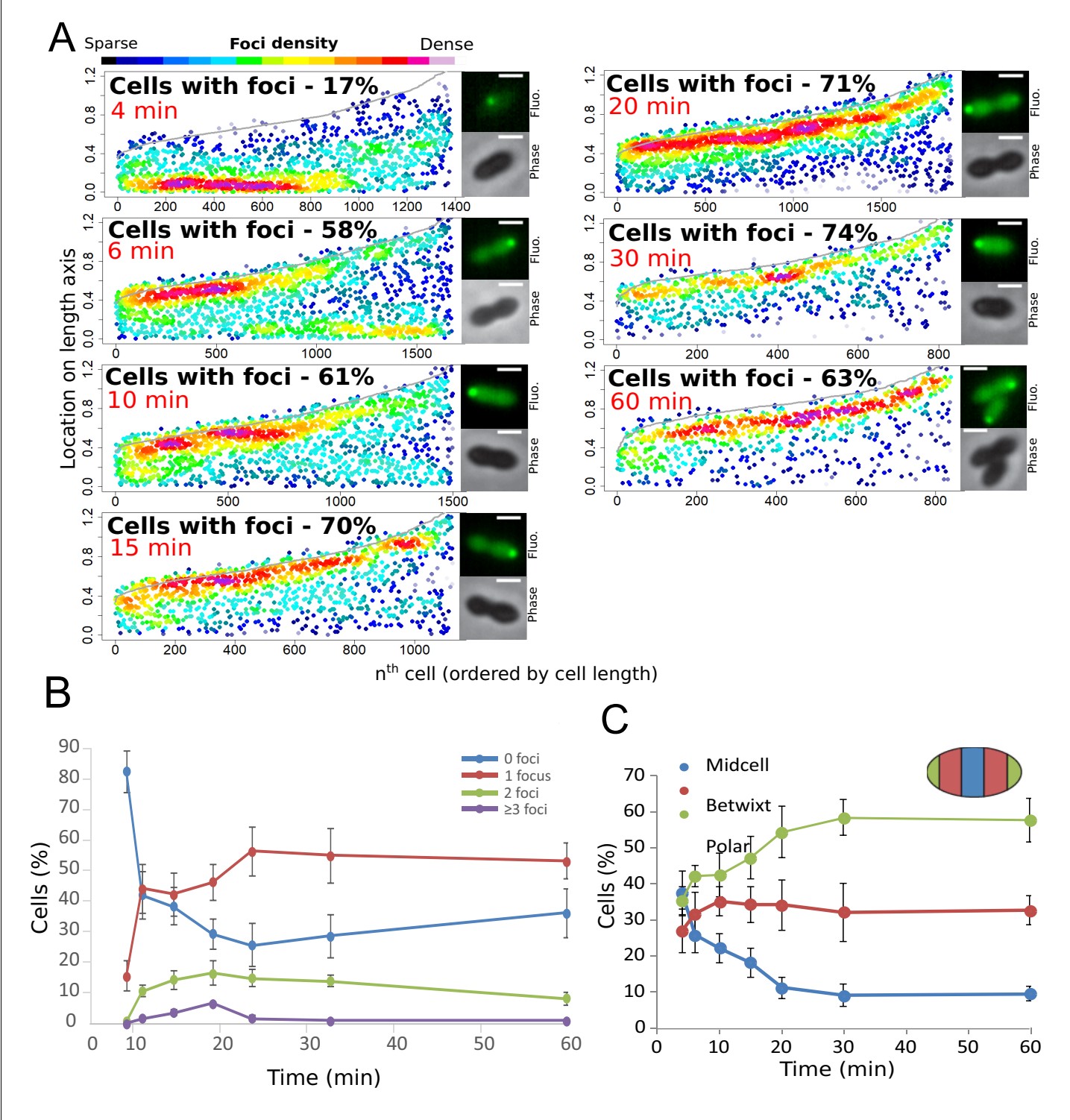

**Figure 3.** Analysis of the cellular localization of DprA-GFP. (**A**) DprA-GFP foci persist at the cell pole after competence shut-off. Data at different time points after CSP addition represented as in *Figure 1E*. 4 min, 8336 cells and 1383 foci analyzed; 6 min, 2871 cells and 1674 foci analyzed; 10 min, 2614 cells and 1502 foci analyzed; 15 min, 1290 cells and 1128 foci analyzed; 20 min, 2110 cells and 1900 foci analyzed; 30 min, 789 cells and 831 foci analyzed; 60 min, 842 cells and 839 foci analyzed. Sample microscopy images of strain R3728 at varying times after competence induction. Scale bars, 1 μm. (**B**) Most competent cells possess a single polar focus of DprA-GFP. Error bars represent triplicate repeats. (**C**) Most cells possess polar DprA-GFP foci. Along the length of a cell of arbitrary length 1, polar foci are found between positions 0–0.15 and 0.85–1, midcell foci are found between 0.35 and 0.65, and anything in between is localized as betwixt. Error bars represent triplicate repeats.

**Video 1.** Timelapse move of DprA-GFP after competence induction by CSP addition. Images taken every 5 min. Left panel, fluorescence; right panel, phase contrast; middle panel, overlay.
https://elifesciences.org/articles/62907#video1

18 mutants, DprA-mKate2 formed foci at levels and localization comparable to wildtype (*Supplementary file 2*). This result contrasted with our previous result (*Figure 4E*), which suggested that expression of the late *com* regulon was required for formation of polar DprA-GFP foci, causing us to revisit our interpretation of *Figure 4C–F*. In fact, to express only the early *com* regulon, we inactivated *comX1*, *comX2*, and *comW* (*Figure 4C*), so this strain produced only early *com* proteins, except σ$^X$ and ComW, and lacked DprA-GFP foci (*Figure 4E*). Conversely, to express only the late *com* regulon, we ectopically expressed *comX* and *comW* (*Figure 4D*), so this strain produced the late *com* regulon but also the early *com* proteins σ$^X$ and ComW and displayed polar DprA-GFP foci at wild-type levels (*Figure 4E and F*). This led us to consider that the only proteins whose presence correlated directly with the presence of polar DprA-GFP foci were thus σ$^X$ and ComW.

To first investigate whether ComW played a role in the formation of polar DprA-GFP foci, the *comW* gene was inactivated in a strain possessing a *rpoD$^{A171V}$* mutation, enabling σ$^X$-RNA polymerase interaction and resulting in late *com* regulon expression in the absence of ComW (*Tovpeko et al., 2016*). DprA-GFP expressed from the native locus still formed polar foci in this strain at levels comparable to the wildtype strain (*Figure 5—figure supplement 1A*). ComW was thus not required for the formation of DprA-GFP foci. In light of this, the only remaining candidate whose presence in competent cells correlated directly with formation of polar DprA-GFP foci was σ$^X$. Thus, to determine if σ$^X$ alone was necessary and sufficient to localize DprA-GFP to the cell poles, both *comX* and *dprA-gfp* or *dprA-gfp* alone were expressed in non-competent *rpoD$^{wt}$* cells. Western blot analysis using α-SsbB antibodies indicated that the late *com* regulon was weakly induced when σ$^X$ was ectopically produced in the absence of *comW* (*Figure 5—figure supplement 1B*). Cells producing DprA-GFP alone showed polar DprA-GFP foci in 10% of cells (*Figure 4G*). In contrast, DprA-GFP foci were formed in 37% of cells when σ$^X$ was also produced in non-competent cells (*Figure 4G*). Importantly, induction of competence in both of these strains resulted in similar foci numbers (*Figure 5—figure supplement 1D*). Altogether, this result suggested that σ$^X$ alone was sufficient to stimulate polar foci of DprA-GFP, highlighting an unexpected role for this early competence σ factor only known to act in concert with ComW to induce late competence gene transcription.

## σ$^X$ mediates the localization of DprA at the cell pole of competent cells

To investigate how σ$^X$ could be involved in the polar localization of DprA-GFP, we explored how it localized in competent cells. To this end, we generated a *comX1-gfp* construct at the native *comX1* locus combined with a wild-type *comX2* gene. In the absence of *comX2*, the second gene producing σ$^X$, σ$^X$-GFP induced the late *com* regulon poorly and was thus weakly transformable (*Figure 5—figure supplement 2*). As a result, a strain possessing *comX1-gfp* and *comX2* was used to determine the localization of σ$^X$-GFP. Remarkably, σ$^X$-GFP formed bright polar foci 15 min after competence induction (*Figure 5A*), reminiscent of those formed by DprA-GFP (*Figure 1C*). Immunofluorescence microscopy on wild-type cells using anti-σ$^X$ antibodies confirmed the polar localization of σ$^X$ in competent cells (*Figure 5—figure supplement 2*). A time-course experiment after competence induction showed that σ$^X$-GFP localized to the cell pole as soon as 4 min after CSP addition (*Figure 5B*), when DprA-GFP forms weak foci at midcell (*Figure 3A*). In contrast, a ComW-GFP fusion protein displayed a diffuse cytoplasmic localization in the majority of competent cells, with only 7% of cells possessing weak foci at the cell poles (*Figure 5—figure supplement 2A* and Supplementary results). Thus, σ$^X$-GFP localizes to the cell pole without its partner in transcriptional activation ComW, despite the fact that σ$^X$ is an alternative sigma factor directing RNA polymerase to specific promoters on the chromosome.

Analysis of σ$^X$-GFP foci distribution showed that they are detected in up to 51% of cells 10 min after competence induction, with most cells possessing a single focus (*Figure 5B and C*). The number of cells with foci decreased steadily after this point (*Figure 5B, C, D*), contrasting with polar

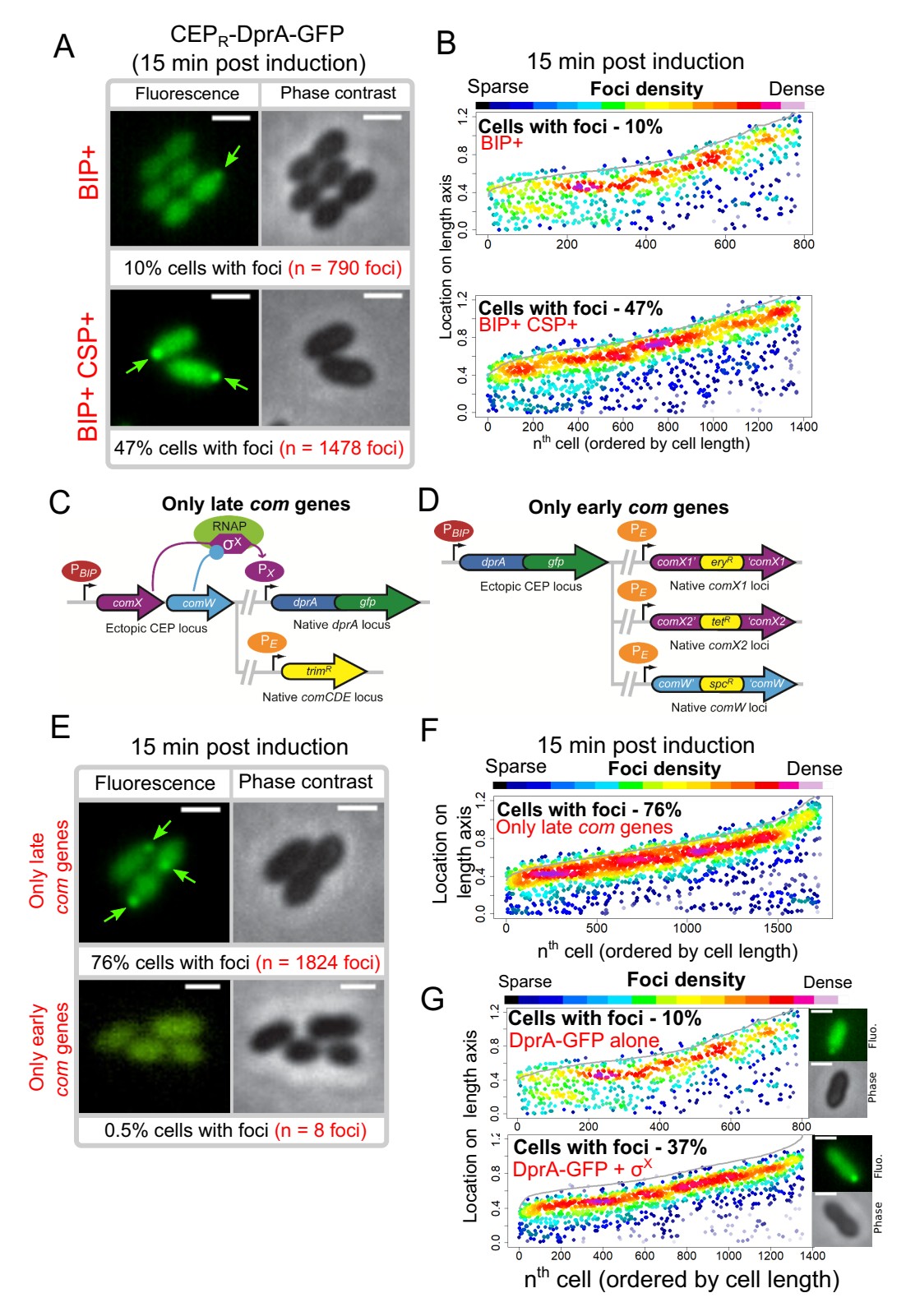

**Figure 4.** Polar accumulation of DprA-GFP appears to depend on late *com* regulon expression. (A) Sample fluorescence microscopy images of strain R4060 producing DprA-GFP 15 min after induction with BIP or BIP and CSP. Scale bars, 1 μm. (B) Competence induction is required for optimal accumulation of DprA-GFP at the cell poles. Focus density maps as in *Figure 1E*. BIP+, 7845 cells and 790 foci analyzed; BIP+ CSP+, 2707 cells and 1478 foci analyzed. (C) Genetic context strain R4107 expressing *dprA-gfp* and only the late *com* regulon. $P_{BIP}$ and $P_X$ as in panel A, $P_E$ represents early

*Figure 4 continued on next page*

*Figure 4 continued*

*com* promoter controlled by ComE. Light blue circle, ComW; light green oval, RNA polymerase; purple hexagon, $\sigma^X$. (D) Genetic context of strain R4140 expressing $CEP_R$-*dprA-gfp* and only the early *com* regulon. $P_{BIP}$ as in panel A, $P_E$ as in panel D. (E) Sample fluorescence microscopy images of strains producing DprA-GFP with only late (R4107) or only early (R4140) *com* operons 15 min after competence induction. Scale bars, 2 μm. (F) Induction of the late *com* regulon is required for accumulation of DprA-GFP at the cell poles. Focus density maps as in *Figure 1E*. 1988 cells and 1824 foci analyzed. (G) Focus density maps produced as in *Figure 1E* from images where DprA-GFP was produced outside of competence in presence or absence of $\sigma^X$. DprA-GFP alone, 7845 cells and 790 foci analyzed; DprA-GPF + $\sigma^X$, 3545 cells and 1355 foci analyzed. Strains used: DprA-GFP alone, R4060; DprA-GFP and $\sigma^X$, R4489.

The online version of this article includes the following figure supplement(s) for figure 4:

**Figure supplement 1.** Validation of strains expressing $CEP_R$-*dprA-gfp* and the late *com* regulon alone.

DprA-GFP which remained stable over 60 min after induction. Importantly, $\sigma^X$-GFP continued to form polar foci in a strain lacking *dprA* (*Figure 5E*), showing that $\sigma^X$ does not depend on DprA for its localization. Together, these results strongly supported the notion that $\sigma^X$ promotes the targeting and assembly of DprA-GFP foci at the cell pole. To further explore this hypothesis, we co-expressed DprA-mTurquoise and $\sigma^X$-YFP fluorescent fusions in the same cells and found that 86% of DprA-mTurquoise foci colocalized with $\sigma^X$-YFP foci (*Figure 5F*). DprA and $\sigma^X$ are thus present at the same pole of the cell at the same time and the polar accumulation of DprA molecules in competent cells depends on $\sigma^X$. These results suggested that $\sigma^X$ could interact with DprA to anchor it to the pole of competent cells. The potential interaction between $\sigma^X$ and DprA was tested in live competent pneumococcal cells in pull-down experiments. To achieve this, we used GFP-TRAP magnetic beads (Chromotek) to purify $\sigma^X$-GFP from competent cells expressing either wild-type DprA or, as a control, the $DprA^{AR}$ mutant that did not accumulate at the cell pole (*Figure 1—figure supplement 1*). Results showed that wild-type DprA co-purified with $\sigma^X$-GFP from competent cells extracts, but $DprA^{AR}$ did not, revealing that $\sigma^X$ and DprA interact in live competent pneumococci (*Figure 5H*). Taken together, these findings reveal that $\sigma^X$ is responsible for the accumulation of DprA at the cell pole during competence.

## Pneumococcal competence induction occurs at the cell pole

We have shown that DprA localization at a single-cell pole correlates with the shut-off of competence (*Figure 2*). Since DprA interacts directly with ComE~P to mediate competence shut-off (*Mirouze et al., 2013*), this raised the question of the subcellular localization of ComD and ComE, which define a two-component system controlling competence regulation. To first explore the localization of ComE in competent cells, a strain was generated expressing a functional *comE-gfp* fluorescent fusion at the native *comE* locus (*Figure 6—figure supplement 1*). In competent cells, ComE-GFP formed patches around the periphery of the cell, often at the cell pole (*Figure 6A, B* and *Figure 6—figure supplement 2A*). A time-lapse experiment showed that these patches were dynamic, navigating around the cell membrane over time (*Video 2*). Next, to explore the sub-cellular localization of ComD, we generated a strain expressing *gfp-comD* at the native *comD* locus (*Figure 1—figure supplement 1A,C*). The resulting GFP-ComD fusion displayed partial functionality in competence induction and transformation (*Figure 1—figure supplement 1D, E*). In contrast to ComE-GFP, GFP-ComD formed distinct polar foci of varying intensity in 57% of cells (*Figure 6A, B*), a localization pattern reminiscent of those observed with DprA-GFP and $\sigma^X$-GFP (*Figures 1* and *5*, respectively). Since GFP-ComD was not fully functional, we also analyzed the localization of a synthetic fluorescent exogenous CSP peptide (CSP-HF, *Figure 6—figure supplement 1F*) in parallel, to track its interaction with ComD at the cell surface. This fluorescent peptide was found to accumulate at a single-cell pole in the majority of competent, wild-type cells (*Figure 6A, B*). In addition, this accumulation was dependent on the presence of ComD (*Figure 6A*), showing that the polar accumulation of the partially functional GFP-ComD fusion represented a functional subcellular localization during competence. In addition, most cells with ComD-GFP foci possessed a single focus (*Figure 6C*), which persisted after the shut-off of competence (*Figure 6—figure supplement 2B*). Altogether, these findings revealed that activation of the positive feedback loop of competence triggered by CSP interaction with ComD to phosphorylate ComE occurs at the cell pole. Because a single pole is mainly targeted by ComD, this also raised the question of its co-localization with ComX and DprA.

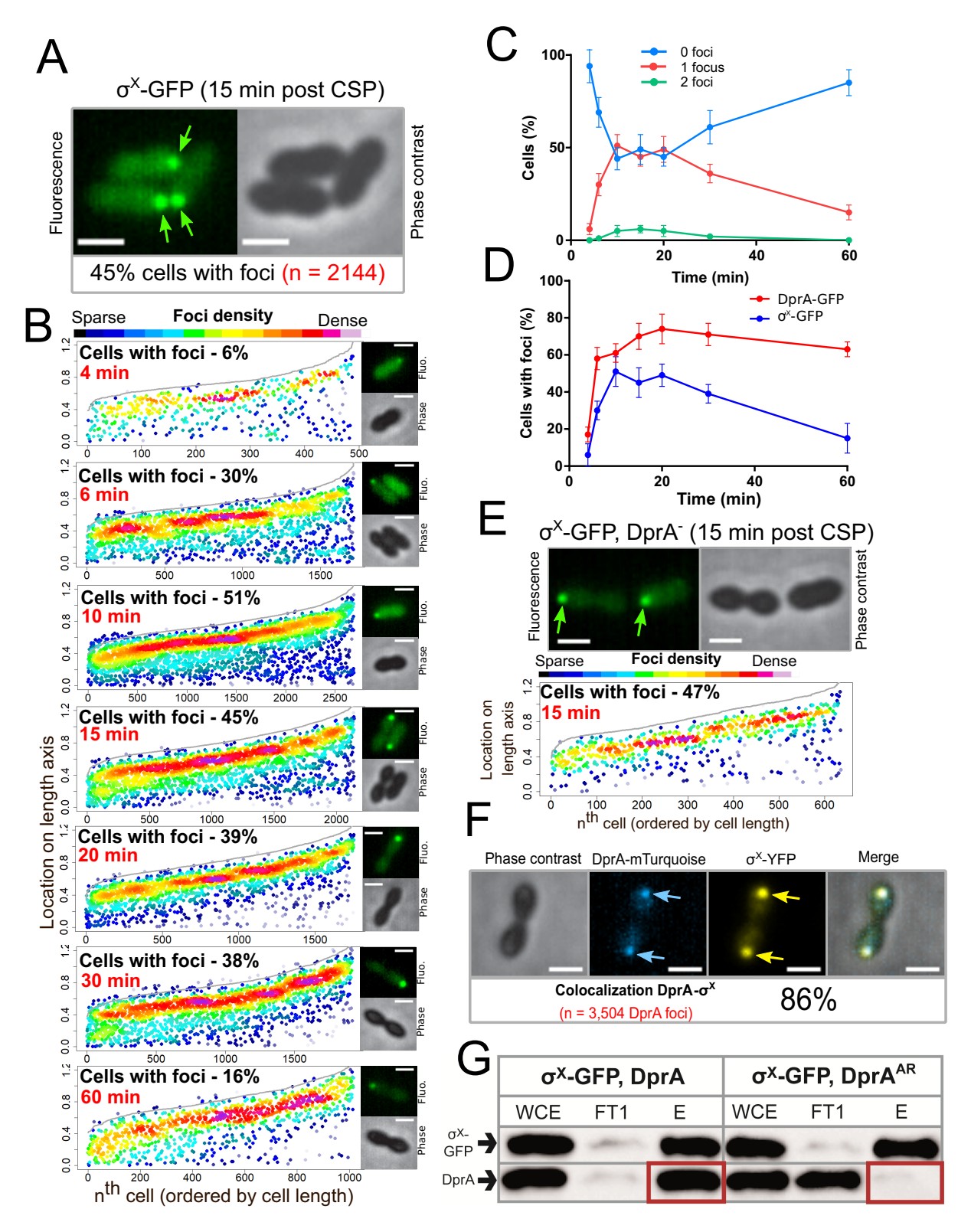

**Figure 5.** σX-GFP interacts directly with DprA at the cell pole during competence. (**A**) Sample fluorescence microscopy images of strain R4451 producing σX-GFP from *comX1* and wild-type σX from *comX2* 15 min after competence induction. Scale bars, 1 μm. (**B**) σX-GFP accumulates at the cell pole during competence. Focus density maps presented as in *Figure 1E*. 4 min, 7544 cells and 489 foci analyzed; 6 min, 5442 cells and 1711 foci analyzed; 10 min, 4358 cells and 2691 foci analyzed; 15 min, 3746 cells and 2144 foci analyzed; 20 min, 4211 cells and 1754 foci analyzed; 30 min, 4695

*Figure 5 continued on next page*

Figure 5 continued

cells and 1920 foci analyzed; 60 min, 5713 cells and 1016 foci analyzed. (C) Most cells have a single $\sigma^X$-GFP focus. Data from the time-course experiment presented in panel B showing the number of foci per cell at each time point. Error bars represent triplicate repeats. (D) DprA-GFP foci persist in cells longer than $\sigma^X$-GFP foci. Comparison of cells with foci at different timepoints from timecourse experiments. DprA-GFP from *Figure 3A*, $\sigma^X$-GFP from panel B. Error bars represent triplicate repeats. (E) Accumulation of $\sigma^X$ at the cell poles does not depend on DprA. Sample microscopy images of a *comX1-gfp*, *dprA⁻* strain (R4469). Focus density maps generated from cells visualized 15 min after competence induction presented as in *Figure 1E*. 1104 cells and 638 foci analyzed. (F) $\sigma^X$ and DprA colocalize at the cell pole. Colocalization of $\sigma^X$-YFP and DprA-mTurquoise in R4473 cells visualized 15 min after competence induction. 7460 cells and 3504 DprA-mTurquoise foci analyzed. Scale bars, 1 μm. (G) DprA is copurified with $\sigma^X$-GFP while DprA$^{AR}$ is not. Western blot of pull-down experiment carried out on strains producing $\sigma^X$-GFP and either DprA (R4451) or DprA$^{AR}$ (R4514) 10 min after competence induction. WCE, whole cell extract; FT1, flow through; E, eluate.

The online version of this article includes the following figure supplement(s) for figure 5:

**Figure supplement 1.** $\sigma^X$ is necessary and sufficient to mediate accumulation of DprA at the cell poles.
**Figure supplement 2.** Validation of *comW* and *comX* fluorescent fusions.

## DprA colocalizes with ComD and CSP during competence

To explore the hypothesis that DprA accumulates at the same pole as ComD to mediate competence shut-off, we performed colocalization analyses of DprA with ComE, ComD, or CSP, respectively, in the same competent cells. Although ComE-YFP foci were dynamic and their localization difficult to analyze, 20% of DprA-mTurquoise foci were nonetheless found to colocalize with ComE-YFP at the cell pole, showing that these proteins can be found at the same pole in the same cells (*Figure 6D*). In contrast, 76% of DprA-mTurquoise foci colocalized with YFP-ComD (*Figure 6E*), showing that these foci form at the same pole in the majority of cells. In addition, 73% of DprA-GFP foci colocalized with CSP-HF (*Figure 6F*). Taken together, these results showed that DprA colocalizes strongly with ComD, generally at one cell pole during pneumococcal competence, further suggesting that the localization of DprA to this cellular location, mediated by $\sigma^X$, facilitates pneumococcal competence shut-off. The localization of DprA at the cell pole where ComD appears to interact with CSP means that DprA is present at the time and place that neophosphorylated ComE~P is produced, allowing DprA to interact with the activated regulator at the cell pole and prevent it from accessing its genomic targets, facilitating shut-off.

## Two copies of *comX* are required for optimal competence shut-off

A distinct hallmark of the single pneumococcal alternative sigma factor $\sigma^X$ of *S. pneumoniae* is that it is produced from two distinct and strictly identical genes, at two distinct loci in the pneumococcal genome, known as *comX1* and *comX2* (*Lee and Morrison, 1999*). However, inactivation of either of these genes had no effect on the efficiency of transformation as previously reported (*Lee and Morrison, 1999*; *Figure 7A*), suggesting that the expression of either *comX1* or *comX2* is sufficient to induce the late *com* regulon to a level ensuring optimal transformation. Having established that $\sigma^X$ plays a second key role in competence shut-off, we explored whether this role required both *comX* genes. Inactivation of either *comX* gene not only slightly reduced the peak of late *com* gene expression but also markedly delayed competence shut-off (*Figure 7B*), revealing that reducing the cellular level of $\sigma^X$ impacts competence shut-off efficiency, presumably because less $\sigma^X$ is present to promote accumulation of DprA at the cell poles. In addition, this unregulated competence shut-off of single *comX* mutants is accompanied by a reduced growth rate of the cell population, consistent with an alteration of cell fitness linked to altered competence shut-off (*Johnston et al., 2018*). In conclusion, this finding suggests that pneumococci possess two copies of *comX* to optimize DprA-mediated competence shut-off and maintain the fitness of competent cells.

## Pre-competence expression of DprA and $\sigma^X$ antagonizes competence induction

This study has uncovered a new functional role of pneumococcal $\sigma^X$ in facilitating DprA-mediated inactivation of ComEP at the cell pole. To obtain further proof that polar DprA plays a role in competence shut-off, we reasoned that if $\sigma^X$ localizes DprA to the cell pole to allow it to interact with neophosphorylated ComE~P, then early ectopic expression of DprA and $\sigma^X$ should antagonize competence induction by interfering with early *com* regulon induction. To test this hypothesis, we

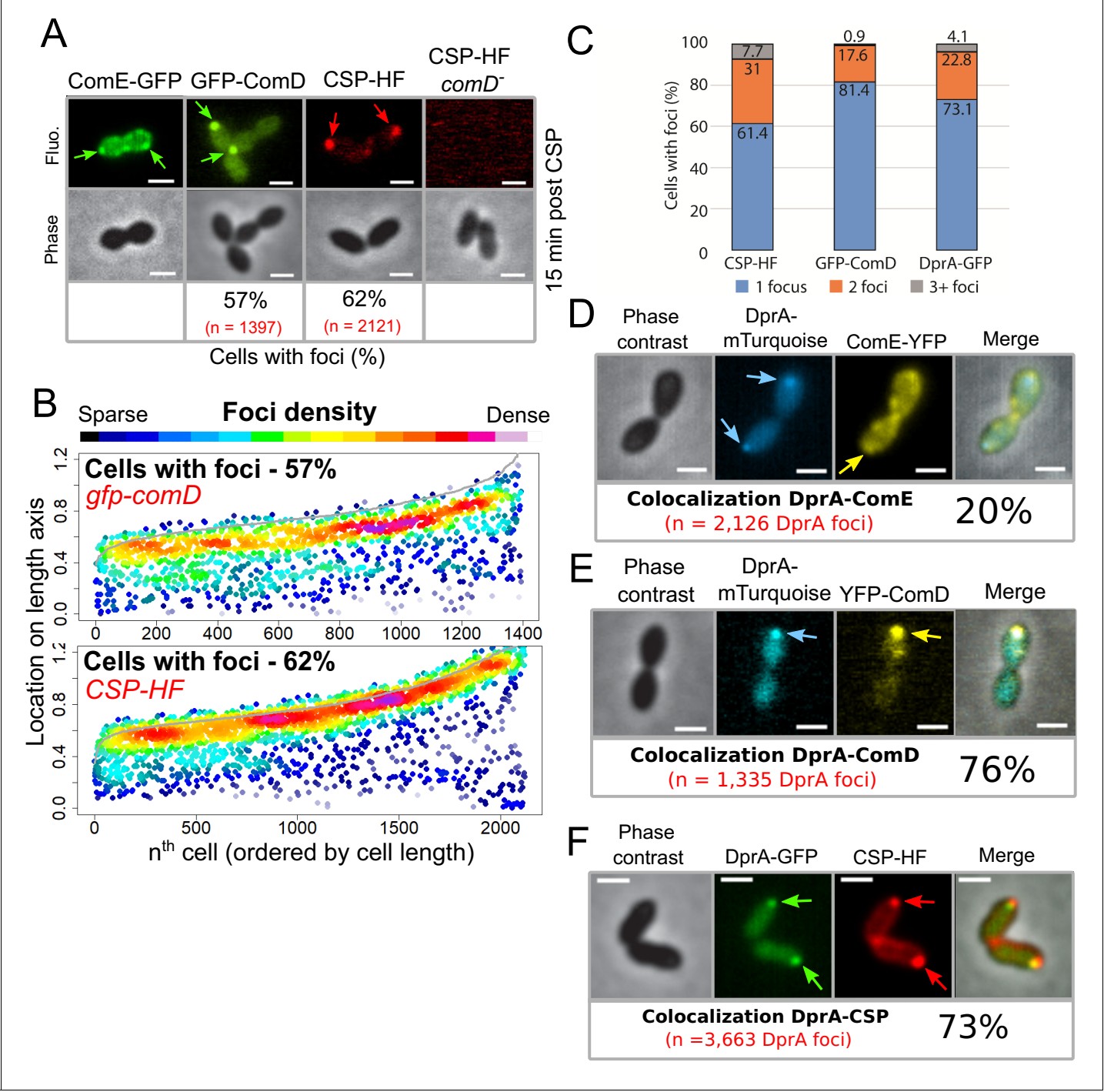

**Figure 6.** The main actors of competence induction and shut-off colocalize at the cell pole. (A) Sample fluorescence microscopy images of cells producing ComE-GFP (R4010), GFP-ComD (R3914) or wildtype cells (R1501) and *comD*⁻ cells (R1745) exposed to CSP-HF. Scale bars, 1 μm. GFP-ComD, 1933 cells and 1397 foci; CSP-HF, 2105 cells and 2121 foci. (B) Focus density maps of cells producing GFP-ComD (R3914) and wild-type (R1501) cells exposed to CSP-HF 15 min after competence induction. Data presented as in *Figure 1E*. GFP-ComD, 1933 cells and 1397 foci analyzed; CSP-HF, 2105 cells and 2121 foci analyzed. (C) Number of foci present in cells possessing foci of different fluorescent fusions 15 min after competence induction. Data taken from panel B except for DprA-GFP, taken from *Figure 3*. (D) DprA-mTurquoise and ComE-YFP colocalization in competent R4176 cells visualized by fluorescence microscopy 15 min after competence induction. 3143 cells and 2126 DprA-mTurquoise foci analyzed. Scale bars, 1 μm. (E) DprA-mTurquoise and YFP-ComD colocalization in competent R4111 cells visualized by fluorescence microscopy 15 min after competence induction. 2857 cells and 1335 DprA-mTurquoise foci analyzed. Scale bars, 1 μm. (F) DprA-GFP and CSP-HF colocalization in competent R4062 cells visualized by fluorescence microscopy 15 min after competence induction. 7588 cells and 3663 DprA-mTurquoise foci analyzed. Scale bars, 1 μm.

*Figure 6 continued on next page*

*Figure 6 continued*

The online version of this article includes the following figure supplement(s) for figure 6:

**Figure supplement 1.** Validation of ComE-GFP, GFP-ComD, and CSP-HF.

**Figure supplement 2.** Time-course experiments tracking the localization of ComE-GFP, GFP-ComD, and CSP-HF after competence induction.

expressed DprA alone or both DprA and σ$^X$ prior to CSP addition to the growth medium (*Figure 7C*), referred hereafter as pre-competence expression. Pre-competence production of DprA alone did not affect competence induction as shown by monitoring luciferase controlled by an early *com* promoter (*Figure 7D*). In these conditions, minimal amounts of DprA accumulated at the cell pole (*Figure 5A*). However, pre-competence expression of both DprA and σ$^X$ resulted in a significantly slower induction of competence (*Figure 7E*). This suggested that σ$^X$-mediated pre-localization of DprA to the cell poles in pre-competent cells, as shown in *Figure 4G*, markedly antagonized CSP induction of competence by allowing DprA to interact with and inactivate neophosphorylated ComE~P. Inactivation of *comW* in this strain to minimize late *com* gene expression did not alter the profiles (*Figure 7F*), showing that the observed effect was directly attributable to the role of σ$^X$ in localizing DprA to the cell poles. Altogether, these findings further prove that in live cells the targeting of pneumococcal DprA to the cell pole, promoted by σ$^X$ anchored at this cellular location, mediates the timely antagonization of the competence induction signal.

## Discussion

### The cell pole defines a competence regulation hub in *S. pneumoniae*

We reported here a spatiotemporal analysis of competence regulation in live pneumococcal cells. We found that the positive regulators ComD, ComE, and σ$^X$, which control the early and late competence expression waves, and the negative regulator DprA colocalize during competence at one cell pole to temporally coordinate its development and shut-off. We have shown that the initial stages of competence induction, relying on the CSP-induced phosphorelay between ComD and ComE, occur at the cell pole (*Figure 7G*). Most importantly, this study uncovered an unexpected second role for σ$^X$ in competence regulation. In addition to controlling the transcription of the late *com* gene *dprA*, σ$^X$ also mediates the accumulation of DprA molecules mostly at the same pole as ComD, a mechanism that facilitates competence shut-off. σ$^X$ thus controls both induction of the late *com* regulon and the shut-off of the early *com* regulon, including thereby its own expression and by consequence that of its regulon. We found that the σ$^X$-directed polar localization of DprA in the vicinity of ComD correlates with its role in competence shut-off. We propose that this targeting favors interaction between DprA and neosynthesized ComE~P to promote efficient shut-off (*Figure 7H*). Importantly, we previously reported that DprA-mediated competence shut-off is crucial to the fitness of competent cells (*Johnston et al., 2018*; *Mirouze et al., 2013*), and we revealed here that this vital role of DprA in actively limiting the competence window is orchestrated at the cell pole, which defines a coordination hub of competence regulation. Unexpectedly, DprA is targeted to this hub by σ$^X$, describing an unprecedented role for an alternative sigma factor.

### The pneumococcal polar competence regulation hub is focused around the histidine kinase ComD

An important finding is the discrete accumulation of the pneumococcal histidine kinase ComD at the cell pole during competence (*Figure 6A, B*). The localization of histidine kinases has not been extensively studied, but few histidine kinases of two-component signaling systems have been found to accumulate at the cell pole. One well-documented example is the *Escherichia coli* histidine kinase CheA, which is involved in chemotaxis. CheA localizes along with its cognate response regulator CheY to a single cell pole,

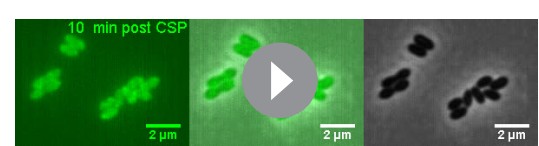

**Video 2.** Timelapse move of ComE-GFP after competence induction by CSP addition. Images taken every 2 min. Left panel, fluorescence; right panel, phase contrast; middle panel, overlay.
https://elifesciences.org/articles/62907#video2

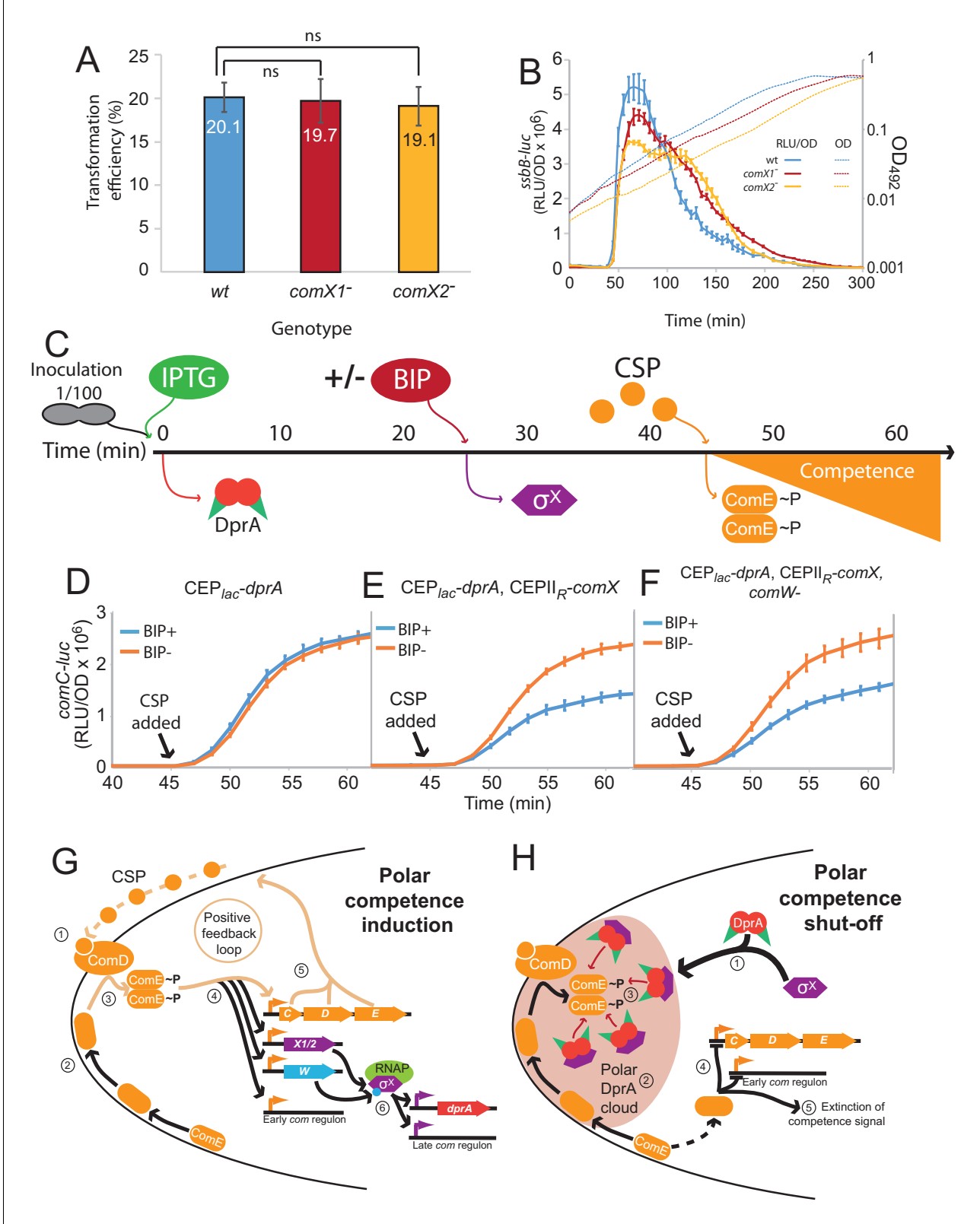

**Figure 7.** Pre-competence production of DprA and σ$^X$ antagonizes competence induction. (**A**) Inactivation of *comX1* or *comX2* does not impact transformation efficiency. Error bars represent triplicate repeats. (**B**) Inactivation of *comX1* or *comX2* delays the shut-off of competence. Data is plotted as Relative Light Units, corrected by optical density (RLU/OD). Error bars represent triplicate repeats. Differences in transformation efficiency of mutant strains determined as non-significant compared to wild-type using GraphPad Prism software (p=0.954 for *comX1$^-$*, p=0.983 for *comX2$^-$*). (**C**) Visual

*Figure 7 continued on next page*

*Figure 7 continued*

representation of experiment exploring the impact of pre-competence production of DprA and σ$^X$ on induction. (**D**) BIP induction of cells lacking CEPII$_R$-*comX* does not alter competence induction. Cells possessing *comC-luc* and CEP$_{lac}$-*dprA* (R4511) were treated as described in Panel C. Error bars represent triplicate repeats. (**E**) Production of σ$^X$ and DprA prior to competence induction antagonizes competence. Cells possessing *comC-luc* and CEP$_{lac}$-*dprA* and CEPII$_R$-*comX* (R4500) were treated as in panel C. Error bars represent triplicate repeats. (**F**) Inactivation of *comW* does not alter the antagonization of competence induction mediated by early DprA and σ$^X$ production. Cells possessing *comC-luc*, CEP$_{lac}$-*dprA*, CEPII$_R$-*comX* and Δ*comW*::*trim* (R4509) were treated as in panel C. Error bars represent triplicate repeats. (**G**) Polar competence induction in *Streptococcus pneumoniae*. (1) Extracellular CSP interacts with ComD at the cell poles, prompting ComD autophosphorylation. (2) Patches of ComE navigate around the cell membrane. (3) Polar ComD phosphorylates ComE. (4) Active ComE~P dimers leave the cell poles to interact with genomic targets, inducting the early *com* regulon and (5) launching an autocatalytic feedback loop. (6) Among the early *com* genes, *comX1*, *comX2*, and *comW* produce σ$^X$ and its activator ComW, which induce the late *com* regulon, including DprA. Orange arrows, early *com* promoters; purple arrows, late *com* promoters. C, *comC*; D, *comD*; E, *comE*; X1/2, *comX1*/*comX2*; W, *comW*; RNAP, RNA polymerase. (**H**) Polar shut-off of pneumococcal competence. (1) σ$^X$ interacts directly with DprA, promoting accumulation of DprA at the cell pole, generating a polar DprA 'cloud' in competent cells, near ComD (2). (3) Polar DprA interacts directly with neosynthesized ComE~P, preventing the regulator from accessing its genomic targets. (4) Unphosphorylated ComE interacts with early *com* promoters, acting as a repressor to prevent induction (*Martin et al., 2013*), promoting extinction of the competence signal, resulting in competence shut-off (5).

forming a chemotactic cluster with a variety of chemoreceptors. CheY stimulates the production of flagellae at the opposite cell pole, driving chemotaxis (*Baker et al., 2006*). Another example is the histidine kinase PilS in *Pseudomonas aeruginosa*, which also accumulates at the cell pole, and along with its cognate cytoplasmic response regulator PilR, regulates expression of polar pili (*Boyd, 2000*). In addition, the PleD response regulator of *Caulobacter crescentus*, which is phosphorylated by both polar DivJ and PleC kinases, localizes to the cell pole and is actively involved in cell cycle control and cell differentiation in *Caulobacter crescentus* (*Paul et al., 2004*). Although ComD tethering at one cell pole is similar to these examples, major differences exist between these two-component systems. Firstly, regarding how their response regulators localize, ComE presents a different localization pattern than CheY, PilR, and PleD, assembling into patches close to the inner side of the cell membrane and focusing dynamically on the cell pole (*Figure 6A* and *Video 2*). ComE phosphorylation by ComD promotes its dimerization and switch into a transcriptional activator (*Martin et al., 2013*). ComE~P dimers should then leave the cell pole to interact with genomic targets and turn on early *com* gene expression, producing σ$^X$ and ComW, which then cooperate to turn on late *com* gene expression. Secondly, the ComDE two-component system induces its own expression, generating a positive feedback loop (*Claverys et al., 2006*; *Martin et al., 2013*), and leading to the strong, rapid induction of competence (*Dagkessamanskaia et al., 2004*; *Peterson et al., 2004*). As found here, this leads to the accumulation of the regulatory proteins at the cell pole hub. In addition, the ComDE two-component system controls a complex, multi-faceted genetic program involving the altered expression of 17% of the genome (*Aprianto et al., 2018*) and is present at a unique cell pole, while the actors of transformation are present at midcell (*Bergé et al., 2013*). Finally, it is possible that the polar localization of the competence regulatory hub is dictated by the very nature of the competence regulation mechanism itself. This localization could be linked to the transient nature of pneumococcal competence. A spatiotemporally controlled localization for this process would provide a mechanism to allow initial induction followed by repression, thus preventing toxicity. It is also possible that the purpose of polar accumulation of σ$^X$ is to limit the circulating levels of σ$^X$. Polar accumulation of σ$^X$ may thus protect the cell from a potentially dangerous hyper-competent state at two levels: firstly by promoting polar DprA accumulation to facilitate shut-off and secondly by sequestering σ$^X$ itself to prevent over-induction of the late *com* regulon. Although the factor localizing σ$^X$ to the cell pole is unknown, it could be considered an anti-sigma factor in this light. Anti-sigma factors can control alternative sigma factors by interacting directly with them to sequester them to the cell membrane to prevent activity until a specific signal is received (*Österberg et al., 2011*). Although ComD goes to the same pole as σ$^X$, it is not the anchor directing σ$^X$ to the pole, since DprA (and thus σ$^X$) still localizes to the cell pole in the absence of early *com* genes (*Figure 4EF*).

## Two copies of *comX* ensure optimal fitness of competent cells

This study has uncovered a second role for σ$^X$ in the shut-off of pneumococcal competence, besides its transcriptional role with ComW, which is to associate with the RNA polymerase and direct the expression of the late *com* regulon. This second role is independent of ComW. It promotes

accumulation of DprA at the cell pole to facilitate competence shut-off. Pneumococci possess two identical copies of *comX* at distinct locations within the genome, called *comX1* and *comX2* (*Lee and Morrison, 1999*). It has remained unclear why two copies exist, since inactivation of a single copy of *comX* does not affect transformation efficiency (*Figure 7A*; *Lee and Morrison, 1999*). The finding of a second role for σ$^X$ in the shut-off of competence suggests a different reason for this duplication, since both copies of *comX* are required for optimal competence shut-off (*Figure 7B*). In light of this, it may be that two copies of *comX* are maintained within the genome to optimize the shut-off of competence. Since unregulated competence is toxic for the cell (*Johnston et al., 2018*), two copies of *comX* provide a fail-safe in case of loss or inactivation of one *comX* gene by transformation or spontaneous mutation, allowing the cell to nonetheless exit the competent state. In light of the second role of σ$^X$ uncovered here, two copies of *comX* provide a simple yet elegant means for *S. pneumoniae* and close relatives to ensure they are always equipped to survive competence, allowing cells to reap the potential benefits of competence without the fitness cost associated with unregulated competence.

## Polar accumulation of DprA and ComD and heterogneity in a post-competence cell population

The σ$^X$-mediated accumulation of DprA in foci at the cell pole depends on a high concentration of DprA molecules in the cell, resulting from the σ$^X$-driven transcription of the *dprA* gene (*Figure 2*). This dual feature implies that polar DprA foci are not formed immediately during competence, in line with our findings (*Figure 3*). We suggest that this short delay provides the opportunity for ComD to phosphorylate ComE and induce competence, before DprA arrives at the cell pole to facilitate shut-off. In addition, the foci formation is dependent on the ability of DprA to dimerize, which alters both its role in transformation via interaction with RecA and its role in competence shut-off via interaction with ComE~P (*Mirouze et al., 2013*; *Quevillon-Cheruel et al., 2012*). However, DprA polar foci formation is independent of the presence of the transformation pore protein ComEC or the recombinase RecA, suggesting that the observed foci are not linked to the conserved role of DprA in transformation (*Figure 1—figure supplement 1*). In addition, the loss of polar foci when reducing cellular levels of DprA-GFP correlates with the loss of competence shut-off, strongly supporting the proposal that DprA accumulation at the cell pole underpins its negative feedback role in competence regulation (*Figure 2* and *Figure 2—figure supplement 1*).

In a competent population, the presence and intensity of DprA-GFP foci varies from cell to cell (*Figure 3*). The same heterogeneity is observed with GFP-ComD foci (*Figure 6—figure supplement 2*). Indeed, at the peak of CSP-induced competence, a quarter of cells do not present detectable DprA-GFP or GFP-ComD foci and, in the other cells, the foci exhibit different level of brightness (*Figure 3* and *Figure 6—figure supplement 2*). In addition, foci are observed in all cell types, meaning that their formation is not determined by a particular stage of the cell-cycle. This highlights a heterogeneity in the pneumococcal competent cell population. Furthermore, DprA-GFP and GFP-ComD foci persist in many cells at least 30 min after the shut-off of competence (*Figure 3* and *Figure 6—figure supplement 2*). It has been previously shown that post-competent cells are unable to respond to CSP for a period of time after they shut-off competence, a phenomenon known as 'blind to CSP' (*Chen and Morrison, 1987*; *Fox and Hotchkiss, 1957*). We suggest that the heterogeneity of polar DprA and ComD accumulation could play a role in this phenomenon. Polar DprA accumulation in a majority of cells could prevent cells from responding to CSP by immediately antagonizing neosynthesized ComE~P, while cells lacking ComD foci may not respond optimally to the competence signal. Our findings explain the suggestion made previously based on a mathematical model simulating competence regulation that a high amount of DprA played a role in this phenomenon (*Weyder et al., 2018*). This notion is supported by the fact that co-expression of DprA and σ$^X$ prior to CSP addition to the cell culture antagonized competence development (*Figure 7C–F*). Furthermore, since not all post-competent cells possess detectable polar foci of DprA-GFP or GFP-ComD, we suggest that whether sufficient DprA or ComD has accumulated at the pole of a particular cell should govern whether this cell can respond to an external CSP signal and is thus receptive to a second wave of competence. This produces a mixture of competent and non-competent cells, which may maximize the potential survival of a pneumococcal population.

## Concluding remarks

In this study, we have shown that the entire pneumococcal competence regulatory cycle occurs at a single cell pole. This generates an asymmetry at the poles of a competent cell, which can be transmitted to future generations and impact the ability to respond to subsequent competence signals. In addition, we have uncovered a key second role for the competence-dedicated alternative sigma factor σ$^X$ that actively localizes DprA to the polar competence regulatory hub to facilitate competence shut-off. This regulatory mechanism, involving two proteins with other conserved roles in competence and transformation respectively, is pivotal to optimal competence shut-off and maintains the fitness of competent cells. This finding represents the first example of an alternative sigma factor playing a central role in the extinction of the signal on which its own production depends and broadens our knowledge of the regulatory roles played by bacterial alternative sigma factors.

# Materials and methods

**Key resources table**

| Reagent type (species) or resource | Designation | Source or reference | Identifiers | Additional information |
|---|---|---|---|---|
| Strain, strain background (*Streptococcus pneumoniae*) | Various | This paper | NBCI Taxon: 1313 | See *Supplementary file 3* |
| Sequence-based reagent | Various oligonucleotides | This paper (Eurofins-MWG) | Primers for cloning | See *Supplementary file 3* |
| Software, algorithm | Fiji | doi:10.1038/nmeth.2019 | RRID:SCR_002285 | |
| Software, algorithm | MicrobeJ | doi: 10.1038/nmicrobiol.2016.77 | www.microbeJ.com | |
| Software, algorithm | R project for statistical computing | http://www.r-project.org/ | RRID:SCR_001905 | |

## Bacterial strains, transformation, and competence

The pneumococcal strains, primers, and plasmids used in this study can be found in *Supplementary file 3*. Standard procedures for transformation and growth media were used (*Martin et al., 2000*). In this study, cells were rendered unable to spontaneously develop competence either by deletion of the *comC* gene (*comC0*) (*Dagkessamanskaia et al., 2004*) or by replacing the *comC* gene which encodes CSP1 with an allelic variant encoding CSP2 (*Pozzi et al., 1996*), since ComD1 is unable to respond to CSP2 (*Johnsborg et al., 2006*; *Weyder et al., 2018*). Both these alterations render cells unable to induce competence using endogenous CSP1. Unless described, pre-competent cultures were prepared by growing cells to an OD$_{550}$ of 0.1 in C+Y medium (pH 7) before 10-fold concentration and storage at –80°C as 100 µL aliquots. Antibiotic concentrations (µg mL$^{-1}$) used for the selection of *S. pneumoniae* transformants were: chloramphenicol (Cm), 4.5; erythromycin (Ery), 0.05; kanamycin (Kan), 250; spectinomycin (Spc), 100; streptomycin (Sm), 200; trimethoprim (Trim), 20. For the monitoring of growth and *luc* expression, precultures were gently thawed and aliquots were inoculated (1 in 100) in luciferin-containing (*Prudhomme and Claverys, 2007*) C+Y medium and distributed (300 ml per well) into a 96-well white microplate with clear bottom. Transformation was carried out as previously described (*Martin et al., 2000*). 100 µL aliquots of pre-competent cells were resuspended in 900 µL fresh C+Y medium with 100 ng mL$^{-1}$ CSP and appropriate IPTG concentrations and incubated at 37°C for 10 min. Transforming DNA was then added to a 100 µL aliquot of this culture, followed by incubation at 30°C for 20 min. Cells were then diluted and plated on 10 mL CAT agar with 5% horse blood before incubation at 37°C for 2 hr. A second 10 mL layer of CAT agar with appropriate antibiotic was added to plates to select transformants, and plates without antibiotic were used as comparison to calculate transformation efficiency where appropriate. Plates were incubated overnight at 37°C. To compare transformation

efficiencies, transforming DNA was either R304 (*Mortier-Barrière et al., 1998*, p.) genomic DNA or a 3434 bp PCR fragment amplified with primer pair MB117-MB120 as noted (*Marie et al., 2017*), both possessing an *rpsL41* point mutation conferring streptomycin resistance. To track competence profiles, a previously described protocol was used (*Prudhomme and Claverys, 2007*). Relative luminescence unit (RLU) and OD values were recorded throughout incubation at 37°C in a Varioskan luminometer (ThermoFisher). The *comC-luc* and *ssbB-luc* reporter genes were transferred from R825 or R895 as previously described (*Berge et al., 2002*; *Chastanet et al., 2001*). $CEP_{lac}$-*dprA-gfp* strains were grown in varying concentrations of IPTG from the beginning of growth, as previously described (*Johnston et al., 2018*).

## Strain construction

Previously published constructs and mutants were simply transferred from published strains by transformation with appropriate selection. The pMB42 plasmid was generated by amplifying *gfp* with primer pair OMB4-OMB5 and the 5′ end of *dprA* with primer pair dprA22-dprA23. The resulting DNA fragments were digested by *Xho*I/*Hind*III and *Xho*I/*Eco*RI, respectively, and both ligated into a pUC59-derived plasmid pAO-0 plasmid digested with *Eco*RI and *Hind*III. This generated a plasmid which, when transformed into pneumococci, generated a strain possessing *dprA-gfp* at the native locus, as well as a *spc* resistance cassette. A strain containing *dprA-gfp* at the native *dprA* locus (R3728) was constructed by transforming R1501 (*comC0*) with the pMB42 plasmid and selecting for spectinomycin resistance. To generate the pMB42-dprA$^{AR}$ and pMB42-dprA$^{QNQ}$ plasmids, the same primer pair was used to amplify *dprA* from strains R2585 and R2830, and the plasmids were generated in the same manner. These plasmids were transformed into R1501 (*comC0*) to generate R4046 and R4047, respectively. To generate pMB42-mKate2, mKate2 was amplified by PCR using the pMK111 plasmid as a template with primer pair CJ431-CJ432. pMB42 plasmid and the resulting PCR product were digested with *Xho*I and *Hind*III restriction enzymes, and ligated together to generate pMB42-mKate2. pMB42-mTurquoise was generated in the same manner using R4011 as template and the CJ454-CJ455 primer pair. These plasmids were transformed into R1501 (*comC0*) to generate R4048 and R4062, respectively.

To generate a strain possessing *comX1-gfp*, a DNA fragment comprising the *gfp* gene flanked by 5′ and 3′ regions of the *comX1* gene was generated by splicing overlap extension (SOE) PCR. The primer pairs used to generate the 5′ *comX1* region, *gfp*, and the 3′ *comX1* region were CJ643-CJ644, CJ645-CJ646, and CJ647-CJ648, using R3728 genomic DNA as a template, respectively. The resulting PCR fragments were fused by SOE PCR and transformed into R1501 (*comC0*) without selection. One hundred clones were screened by PCR with the primer pair CJ643-CJ648 to identify *comX1-gfp* clones, one of which was named R4451. To generate a strain possessing *comE-gfp*, the same method was used with the primer pairs CJ379-CJ380, CJ381-CJ382, and CJ383-CJ384 to generate the individual fragments and CJ379-CJ384 to generate the SOE fragment. To generate a strain possessing *comW-gfp*, the same method was used with the primer pairs CJ508-CJ514, CJ515-CJ516, and CJ517-CJ513 to generate the individual fragments and CJ508-CJ513 to generate the SOE fragment. To generate *gfp-comD*, PCR fragments of the regions up and downstream of the *comD* start codon and *gfp*, by PCR on R3728 with primer pairs MP216-MP217, MP220-MP221, and MP218-MP218, respectively. SOE PCR on these three fragments using primer pair MP216-MP221 generated *gfp-comD* with flanking sequences. This fragment was transformed into R1036 (*rpsL1*, Δ*comC::kan-rpsL*), which contains a Janus cassette in the *comCDE* locus (*Sung et al., 2001*), with streptomycin selection, to replace the Janus cassette as previously described (*Sung et al., 2001*). To generate the plasmid pCEP$_R$-dprA-gfp, *dprA-gfp* was amplified from R3728 using primer pair CJ410-CJ411, possessing restriction sites *Nco*I and *Bam*HI. This fragment and the pCEP$_R$-luc (*Johnston et al., 2016*) plasmid were digested and ligated together to generate pCEP$_R$-dprA-gfp. This plasmid was transformed into R1501 (*comC0*) with kanamycin selection to generate R4045. For each fluorescent fusion, a LEGSG linker was placed between *gfp* and the gene as previously described (*Bergé et al., 2013*).

To inactivate the 18 *cin* boxes of each late *com* operon individually, two PCR fragments were generated for each with homology to the 5′ and 3′ regions flanking the *cin* boxes using primers DDL48 to DDL121 (See *Supplementary file 3* for details), with the entire 8 bp *cin* box (TACGATAA, *Lee and Morrison, 1999*) replaced with a *Bam*HI site and two flanking base pairs as shown (A*GGATCC*T). In some cases, the first and last bases were altered so as not to match those of a

particular *cin box* sequence (*Supplementary file 3*). SOE PCR using the flanking primers for each *cin* box (e.g. DDL64 and DDL67 for *dprA^cinbox-^*) generated a single fragment of ~ 2500 bp with the mutated *cin* box flanked by ~ 1250 bp of homology on either side. These SOE PCR fragments were individually transformed into R4431 (*comC0, radA-gfp, dprA-mKate2*), and 20 clones were picked for each transformation. The *radA-gfp* construct was present in this strain as part of another study and will not be discussed here, but explains why the *dprA-mKate2* fusion was used here. The targeted *cin* box sites were amplified by PCR on 10 clones in each case, using the same primers as for the SOE PCR, then digested by *Bam*HI and screened on agarose gel (1%). Mutated clones were identified as those whose amplified PCR fragment was digested by *Bam*HI, indicating insertion of the transforming DNA fragment and resulting *cin* box mutation.

To generate the pCEP$_{lac}$-dprA-gfp plasmid, $P_{syn}$-lacI-$P_{lac}$-dprA and $P_{syn}$-kan-treR fragments were amplified from the CEP platform in strain R3833 using primer pairs AmiF1-CJ496 and CJ499-CJ500, while *gfp* was amplified from R3728 using primer pair CJ497-CJ498. These three fragments were used to generate a large CEP$_{lac}$-dprA-gfp SOE PCR fragment using primer pair AmiF1-CJ500, possessing the entire CEP$_{lac}$-dprA-gfp platform as well as flanking sequence, which was then transformed into R1501 (*comC0*) with kanamycin selection to generate R4261.

To delete *comW* and replace it with a trimethoprim resistance cassette, the sequences 5′ and 3′ of the *comW* gene were amplified using primer pairs CJ559-CJ560 and CJ563-CJ564, and the *trim* resistance cassette was amplified from strain TK108 (*Kloosterman et al., 2006*) by primer pair CJ561-CJ562. These three fragments were fused by SOE PCR with the primer pair CJ559-CJ564 and the resulting fragment was transformed into R1501 (*comC0*) to generate R4575 (*comC0, ΔcomW::trim*). To delete the entire *comCDE* operon and replace it with a trimethoprim resistance cassette, the sequences 5′ and 3′ of the *comW* gene were amplified using primer pairs CJ385-CJ470 and CJ473-CJ384, and the *trim* resistance cassette was amplified from strain TK108 (*Kloosterman et al., 2006*) by primer pair CJ471-CJ472. These three fragments were fused by SOE PCR with primer pair CJ385-CJ384 and the resulting fragment was transformed into R1501 (*comC0*) to generate R4574 (*ΔcomCDE::trim*).

## Pre-competence expression of DprA and σ$^X$

To test whether pre-competence expression of DprA and σ$^X$ antagonized competence induction, we generated strains which possessed an ectopic platform expressing *dprA* in response to IPTG (CEP$_{lac}$-dprA) (*Johnston et al., 2018*) alone or in addition to an ectopic platform expressing *comX* (CEPII$_R$-comX) in response to the BIP peptide, a peptide controlling a two-component system with an induction profile similar to CSP (*de Saizieu et al., 2000*; *Johnston et al., 2016*). We showed previously that ectopic expression of *dprA-gfp* and *comX* resulted in localization of DprA-GFP to the cell poles, unlike ectopic expression of *dprA-gfp* alone (*Figure 5A*). In the present experiment, expression of *dprA* was induced throughout growth by the presence of 50 µM IPTG in the culture medium. Cells were grown to OD$_{550}$ 0.08, and were either exposed to BIP to induce *comX*, or not. Twenty minutes after this, CSP was added to the culture to induce competence, and the induction of the early *com* regulon was tracked by measuring activity of a *comC-luc* transcriptional reporter fusion (*Berge et al., 2002*) every 2 min for 20 min to observe initial competence induction.

## Fluorescence microscopy and image analysis

Pneumococcal precultures grown in C+Y medium at 37°C to an OD$_{550}$ of 0.1 were induced with either CSP (100 ng mL$^{-1}$) or BIP (250 ng mL$^{-1}$) peptide. At indicated times post-induction, 1 mL samples were collected, cooled down by addition of 500 mL cold medium, pelleted (3 min, 3000 g), and resuspended in 1 mL C+Y medium. Of this suspension, 2 µL were spotted on a microscope slide containing a slab of 1.2% C+Y agarose as previously described (*Bergé et al., 2013*) before imaging. Unless stated, images were visualized 15 min after competence induction, at the peak of competence gene expression. To generate videos, images were taken of the same fields of vision at varying time points during incubation at 37°C. Images were captured and processed using the Nis-Elements AR software (Nikon). Images were analyzed using MicrobeJ, a plug-in of ImageJ (*Ducret et al., 2016*). Data was analyzed in R and unless stated, represented as focus density maps plotted on the longitudinal axis of half cells ordered by cell length. Each spot represents the localization of an individual focus, and spot color represents focus density at a specific location on the half cell. Cells

with > 0 foci shown for each time point. In cells possessing > 1 foci, foci were represented adjacently on cells of the same length.

## Western blots

To compare the expression profiles of competence proteins after competence induction, time-course western blots were carried out. Cells were diluted 100-fold in 10 mL C+Y medium pH 7 and grown to OD 0.1. Where appropriate, cells were induced with CSP (100 ng mL$^{-1}$) or BIP (250 ng mL$^{-1}$) peptide. At indicated time points, OD$_{550}$ measurements were taken and 500 μL of culture was recovered. Samples were centrifuged (3 min, 3000 g) and pellets were resuspended in 40 μL of TE 1x supplemented with 0.01% DOC and 0.02% SDS. Samples were then incubated for 10 min at 37°C before addition of 40 μL 2x sample buffer with 10% β-mercaptoethanol, followed by incubation at 85°C for 10 min. Samples were then normalized compared to the initial OD$_{550}$ reading, and loaded onto SDS-PAGE gels (BIORAD). Samples were migrated for 30 min at 200V, and transferred onto nitrocellulose membrane using a Transblot Turbo (BIORAD). Membranes were blocked for 1 hr at room temperature in 1x TBS with 0.1% Tween20% and 10% milk, before two washes in 1x TBS with 0.1% Tween20 and probing with primary antibodies (1/10,000 as noted) in 1x TBS with 0.1% Tween20% and 5% milk overnight at 4°C. After a further four washes in 1x TBS with 0.1% Tween20, membranes were probes with anti-rabbit secondary antibody (1/10,000) for 1 hr 30 min, followed by another four washes in 1x TBS with 0.1% Tween20. Membranes were activated using Clarity Max ECL (BIORAD) and visualized in a ChemiDoc Touch (BIORAD).

## Construction and validation of strains expressing only early or late *com* regulons

To express only the late *com* regulon, the *comX* and *comW* genes were placed under the control of BIP at the CEP expression platform (*Weyder et al., 2018*) in a strain possessing *dprA-gfp* at the native locus, with the *comCDE* operon deleted to prevent any expression of early *com* operons (*Figure 4C*). In addition, the *cbpD* gene was inactivated to compensate for the absence of the early com gene *comM* (*Bergé et al., 2017*; *Guiral et al., 2005*). Addition of BIP to this strain resulted in induction of the late *com* regulon in the absence of the autocatalytic feedback loop generated by the early *com* regulon. This strain transformed equally well upon CSP or BIP addition, showing that late *com* regulon expression was sufficient for transformation (*Figure 4—figure supplement 1*). However, since the catalytic feedback loop of the early *com* genes was removed, little to no shut-off of late *com* gene expression was observed (*Figure 4—figure supplement 1*). To express only the early *com* regulon, both copies of *comX*, along with *comW*, were inactivated in a strain possessing CEP$_R$-*dprA-gfp* (*Figure 4D*). Addition of both BIP and CSP to this strain resulted in the induction of the early *com* regulon and the production of DprA-GFP. As no other late *com* genes were expressed in this strain, the strain could not transform and the induction profile of *ssbB-luc* could not be tracked.

## Immunofluorescence microscopy

To visualize native DprA and σ$^X$ in live competent pneumococci, immunofluorescence microscopy was carried out as described previously (*Wayne et al., 2010*), with alterations. Briefly, cells were grown in 3 mL C+Y medium at 37°C until OD$_{550}$ 0.1. Competence was induced by addition of CSP (100 ng mL$^{-1}$) and cells were incubated for 15 min at 37°C before recovery of cell pellets by centrifugation. Pellets were fixed by resuspending in 4% formaldehyde (Sigma F8775) in PBS and incubation for 15 min at room temperature followed by 45 min on ice. Cells were then washed three times in 1 mL PBS and resuspended in 0.3 mL chilled GTE (50 mM glucose, 20 mM Tris-HCl pH 7.5, 10 mM EDTA). 50 μL of cells were transferred onto polylysine slides (Sigma P0425) and incubated for 5 min at room temperature before aspiration of excess liquid, with slides then washed twice with 50 μL PBS. 0.2% Triton X100 in PBS (PBS-T) was added for 10 s before being aspirated, and slides were dipped in methanol at −20°C for 10 min and air-dried. Slides were then rehydrated with 50 μL PBS, and incubated with 50 μL PBS-T for 5 min at room temperature before a 1 hr incubation with 50 μL 5% skimmed milk in PBS-T (PBS-T-M). After this, slides were washed twice with 50 μL PBS and incubated with 50 μL of 1/100 dilutions of polyclonal primary antibodies raised in rabbits against DprA and σ$^X$ in PBS-M for 1 hr at room temperature. Slides were washed twice in 50 μL PBS for 10 s and

once for 5 min, before incubation with 50 µL of 1/100 goat anti-rabbit IgG coupled to Alexa Fluor 488 (Invitrogen A32731) with 0.2 µg mL$^{-1}$ 4',6-Diamidino-2-phenylindole dihydrochloride (DAPI, Sigma D9542) in PBS-M for 1 hr at room temperature in the dark. Slides were then washed twice in 50 µL PBS for 10 s and once for 5 min before addition of 10 µL VectaShield mounting solution (VectorLabs H-2000) and visualization under the microscope.

### Co-immunoprecipitation

Co-immunoprecipitation was done using magnetic GFP-Trap beads as per manufacturer's instructions (Chromotek). Briefly, cells were inoculated 1/100 in 25 mL of C+Y medium pH 7 and grown to OD$_{550}$ 0.1. Competence was induced by addition of 100 ng mL$^{-1}$ CSP, and cells were incubated for 10 min at 37°C. Cultures were mixed with 25 mL cold buffer A (10 mM Tris pH 7.5, 150 mM NaCl) and centrifuged for 15 min at 5000 g. Pellets were washed twice with 10 mL cold buffer A and stored at −80°C until use. After defrosting, pellets were resuspended in 1 mL buffer B (10 mM Tris pH 7.5, 150 mM NaCl, 0.2 mM EDTA, 0.1% TritonX100, 1 M DTT) and incubated for 10 min on ice, followed by 10 min at 37°C, and a further 10 min on ice. Samples were then sonicated (2 × 30 s with 10 s pause) and centrifuged for 30 min at 4°C and 16,000 g. After normalizing the protein concentrations in the samples, 20 µg mL$^{-1}$ RNAse A and 50 µg mL$^{-1}$ DNAse I were added and samples were tumbled end over end at 4°C for 30 min. 75 µl of GFP-Trap beads were added to the samples, which were then tumbled end over end at 4°C for 2 hr 30 min. GFP-Trap beads were purified by magnetism and washed twice in 500 µL ice cold dilution buffer (10 mM Tris pH 7.5, 150 mM NaCl, 0.5 mM EDTA), before being resuspended in 2x sample buffer + 10% β-mercaptoethanol and incubated at 95°C for 10 min. Samples were then run on SDS-PAGE gel and western blots carried out as described above.

### Pre-competence expression of DprA and σ$^X$

Cells (R4500, CEP$_{lac}$-dprA, dprA::spc, CEPIIR- R4509, R4511,) were grown to OD492 0.2 in 2 mL C+Y medium (pH7.6) with 50 µM IPTG. After centrifugation for 5 min at 5000 rpm, cells were resuspended in 1 mL C+Y medium and stored at −80°C in 100 µL aliquots until required. Aliquots were resuspended in 900 µL fresh C+Y medium (pH 7.6) and diluted 1/10 in a 96-well plate in C+Y medium (pH 7.6) with luciferin and 50 µM IPTG to induce DprA expression. After 25 min, BIP (250 ng µL$^{-1}$) was added where noted to induce σ$^X$. Twenty minutes later, CSP (100 ng µL$^{-1}$) was added to induce competence for 20 min. Luminometric and photometric reading were taken every 2 min during this time to report induction of comC-luc.

## Acknowledgements

We thank Nathalie Campo and Mathieu Bergé for critical reading of the manuscript, and the rest of the Polard lab for helpful discussions. We thank Jérôme Rech for help creating Videos. We thank Jan-Willem Veening for kind donation of the pMK111 plasmid. This work was funded by the Agence Nationale de la Recherche (Grants ANR-10-BLAN-1331 and ANR-13-BSV8-0022).

## Additional information

### Funding

| Funder | Grant reference number | Author |
| --- | --- | --- |
| Agence Nationale de la Recherche | ANR-10-BLAN-1331 | Patrice Polard |
| Agence Nationale de la Recherche | ANR-13-BSV8-0022 | Patrice Polard |

The funders had no role in study design, data collection and interpretation, or the decision to submit the work for publication.

## Author contributions
Calum HG Johnston, Conceptualization, Data curation, Formal analysis, Investigation, Methodology, Writing - original draft, Writing - review and editing; Anne-Lise Soulet, Matthieu Bergé, Investigation; Marc Prudhomme, Conceptualization, Investigation, Writing - review and editing; David De Lemos, Data curation, Investigation; Patrice Polard, Conceptualization, Formal analysis, Supervision, Funding acquisition, Investigation, Writing - review and editing

## Author ORCIDs
Calum HG Johnston https://orcid.org/0000-0001-5942-9117
Matthieu Bergé http://orcid.org/0000-0002-0910-6114
Patrice Polard https://orcid.org/0000-0002-0365-4347

## Decision letter and Author response
Decision letter https://doi.org/10.7554/eLife.62907.sa1
Author response https://doi.org/10.7554/eLife.62907.sa2

## Additional files
### Supplementary files
• Supplementary file 1. Validation of *cin box* mutant inactivations. Successful *cin box* inactivation was confirmed in three individual *cin box* mutants with known transformation deficits, and each strain displayed the expected transformation deficit, showing that as expected, full *cin box* mutation successfully abrogated $\sigma^X$-mediated expression, and allowing extrapolation to the 18 *cin box* mutants generated.

• Supplementary file 2. Effect of *cin box* inactivations on the presence of polar DprA-mKate2 foci.

• Supplementary file 3. Strains, plasmids and primers used in this study. [a] Antibiotic resistances: $Spc^R$, spectinomycin; $Kan^R$, kanamycin; $Trim^R$, trimethoprim; $Sm^R$, streptomycin; $Cm^R$, chloramphenicol; $Rif^R$, rifampicin; $Nov^R$, novobiocin; $Ery^R$, erythromycin; $Tet^R$, tetracycline; $Amp^R$, ampicillin. [b] Underlined, italic bases represent restriction sites used for cloning. References cited in this file: *Campbell et al., 1998*; *Burghout et al., 2007*; *Desai and Morrison, 2006*; *Attaiech et al., 2008*; *Morrison et al., 2007*; *Martin et al., 1995*; *Sanchez-Puelles et al., 1986*; *Laurenceau et al., 2013*; *Laurenceau et al., 2015*; *Diallo et al., 2017*; *Lefevre et al., 1979*; *Caymaris et al., 2010*; *Mortier-Barrière et al., 2019*; *Akerley et al., 1998*; *van Raaphorst et al., 2017*; *Guiral et al., 2006*.

• Transparent reporting form

## Data availability
All data generated or analysed during this study are included in the manuscript and supporting files.

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
