## [Decision Letter]

**Acceptance summary:**

In your study, you investigated competence proteins of the human pathogen *Streptococcus pneumoniae* and discovered interesting new localization pattern. Notably, you provided evidence that the competence-specific sigma factor SigmaX recruits the DprA protein to the cell pole, which, ultimately, leads to competence termination. By putting this additional spatial layer onto the well-described temporal expression pattern, this study provides important new insight into competence regulation in the pneumococcus.

**Decision letter after peer review:**

Thank you for submitting your article "The alternative σ factor σX mediates competence shut-off at the cell pole in *Streptococcus pneumoniae*" for consideration by *eLife*. Your article has been reviewed by three peer reviewers, including Melanie Blokesch as the Reviewing Editor and Reviewer #1, and the evaluation has been overseen by Gisela Storz as the Senior Editor. The following individual involved in review of your submission has agreed to reveal their identity: David Dubnau (Reviewer #3).

The reviewers have discussed the reviews with one another and the Reviewing Editor has drafted this decision to help you prepare a revised submission.

Summary:

This study investigates natural competence regulation in *Streptococcus pneumoniae* with a focus on competence shut-off. While previous work by the group had shown that the DNA processing protein A (DprA) was involved competence termination, its mode-of-action was not fully understood. Here, the protein's mode-of-action was addressed by adding a spatial regulation layer on top of the temporal competence window. Precisely, the authors provided evidence that DprA is recruited to the cell pole and that this localization is important to turn off the competence program. Moreover, DprA's localization was shown to be stimulated by the accumulation of the comX-encoded σ factor SigX at the same pole and a direct interaction between these proteins was demonstrated. The polar location of DprA is thought to inactivate the response regulator (ComE~P), a process that ultimately ended the competence induction circle. Collectively, these data provide new insights into the spatiotemporal choreography of competence proteins and how certain protein relocalization events to the pole are required to switch off competence in this organism. The finding are especially remarkable given the long-known roles of SigX and DprA in competence induction and DNA recombination mediation, respectively, while their role in competence shut-off was mostly understudied in the past.

Essential revisions:

All three experts agreed that this study is very well performed, that it includes important controls, and that the manuscript is clearly written. The only minor concern that the experts had was the impact of the fused GFP part on SigX1's localization pattern. Indeed, with GFP being prone to dimerization, the fusion protein might not fully reflect the localization of the unfused σ factor. While the localization dynamics of SigX1-GFP somewhat speaks against an artifact, the experts wonder if the authors could do or have tried immunofluorescence microscopy on native SigX or a modified version with a short detection tag? Such experiments would further strengthen the manuscript. In case such experiments are technically unfeasible, the authors should nonetheless mention this caveat in the text of their manuscript.

In this context, we would like to draw your attention to changes in our revision policy that we have made in response to COVID-19 (https://elifesciences.org/articles/57162). Specifically, we are asking editors to accept without delay manuscripts, like yours, that they judge can stand as *eLife* papers without additional data, even if they feel that they would make the manuscript stronger.

---

## [Author Response]

Essential revisions:All three experts agreed that this study is very well performed, that it includes important controls, and that the manuscript is clearly written. The only minor concern that the experts had was the impact of the fused GFP part on SigX1's localization pattern. Indeed, with GFP being prone to dimerization, the fusion protein might not fully reflect the localization of the unfused σ factor. While the localization dynamics of SigX1-GFP somewhat speaks against an artifact, the experts wonder if the authors could do or have tried immunofluorescence microscopy on native SigX or a modified version with a short detection tag? Such experiments would further strengthen the manuscript. In case such experiments are technically unfeasible, the authors should nonetheless mention this caveat in the text of their manuscript.In this context, we would like to draw your attention to changes in our revision policy that we have made in response to COVID-19 (https://elifesciences.org/articles/57162). Specifically, we are asking editors to accept without delay manuscripts, like yours, that they judge can stand as eLife papers without additional data, even if they feel that they would make the manuscript stronger.

In response to the main point regarding the localization of σ^X^-GFP to the cell pole, we have carried out an immunofluorescence control looking at σ^X^ specifically, using anti-σ^X^ antibodies. This was done previously for DprA in Figure 1F. The results show that σ^X^ accumulates at the cell pole in native conditions, confirming the relevance of the localization of the σ^X^-GFP fusion. This information has been added to the subsection “σ X mediates the localization of DprA at the cell pole of competent cells” and to Figure 5—figure supplement 2. We also added an immunofluorescence microscopy section to the Materials and methods section, as this was absent. We thank the reviewers for this suggested control, which strengthens the arguments of our paper.